# Diversified Flow Matching with Translation Identifiability

Sagar Shrestha [1]   Xiao Fu [1]

## Abstract

Diversified distribution matching (DDM) finds a unified translation function mapping a diverse collection of conditional source distributions to their target counterparts. DDM was proposed to resolve content misalignment issues in unpaired domain translation, achieving translation identifiability. However, DDM has only been implemented using GANs due to its constraints on the translation function. GANs are often unstable to train and do not provide the transport trajectory information—yet such trajectories are useful in applications such as single-cell evolution analysis and robot route planning. This work introduces *diversified flow matching* (DFM), an ODE-based framework for DDM. Adapting flow matching (FM) to enforce a unified translation function as in DDM is challenging, as FM learns the translation function's velocity rather than the translation function itself. A custom bilevel optimization-based training loss, a nonlinear interpolant, and a structural reformulation are proposed to address these challenges, offering a tangible implementation. To our knowledge, DFM is the first ODE-based approach guaranteeing translation identifiability. Experiments on synthetic and real-world datasets validate the proposed method.

## 1. Introduction

*Unpaired domain translation* (UDT) aims to translate samples from one domain to another (e.g., photographs to cartoons) while keeping the high-level semantic meaning (or "content"). Here, "unpaired" means that the translation is done without using paired cross-domain samples. UDT has achieved significant empirical successes in various applications, such as unpaired image-to-image translation (Zhu et al., 2017; Choi et al., 2018; Huang et al., 2018; Yang

---

[1]School of Electrical Engineering and Computer Science, Oregon State University, Corvallis, Oregon, USA. Correspondence to: Xiao Fu <xiao.fu@oregonstate.edu>.

*Proceedings of the $42^{nd}$ International Conference on Machine Learning*, Vancouver, Canada. PMLR 267, 2025. Copyright 2025 by the author(s).

et al., 2023), medical imaging (Kong et al., 2021; Song et al., 2024), and single-cell data analysis (Tong et al., 2023; Kapuśniak et al., 2024).

UDT is commonly realized by transporting the distribution of the source domain to that of the target domain (see, e.g., (Zhu et al., 2017; Liu et al., 2023b)). Distribution transport is a core task in modern machine learning. It is heavily studied in the context of domain adaptation, transfer learning, and generative models. Distribution transport can be realized by popular tools such as MMD (Long et al., 2016), GANs (Goodfellow et al., 2014), Shrödinger Bridge (De Bortoli et al., 2021), and continuous normalizing flows (CNF) (Lipman et al., 2023). However, it has been noted that UDT often loses control of the content to be produced in the target domain. For example, in writing style translation, a handwritten digit "7" could be translate to printed "3" under perfect distribution transport (Moriakov et al., 2020; Shrestha & Fu, 2024). Similar issues are seen in Fig. 1, where source pictures are translated to cartoons of unrelated persons. This "content misalignment" issue arises as UDT does not have identifiability of the intended translation function (i.e., the function that translates handwritten "7" to printed "7" and the function that converts profile pictures to cartoon faces without losing identities). There could be an infinite number of translation functions that can attain perfect distribution transport among the domains (Moriakov et al., 2020; Shrestha & Fu, 2024).

Many attempts have been made to address the content misalignment issue; see, e.g., (de Bézenac et al., 2019; Zhu et al., 2017; Taigman et al., 2017; Liu et al., 2017; Xu et al., 2022) for various regularization strategies to empirically enforce translation identifiability. Notably, the recent work (Shrestha & Fu, 2024) proposed the so-called *diversified distribution matching* (DDM) approach, which showed that UDT can be made identifiable if the translation function is learned from simultaneously transporting a number of diverse conditional distribution pairs.

**Challenges.** The DDM criterion provides a theory-driven approach to avoid translation non-identifiability. However, similar as many works in this domain (e.g., (Zhu et al., 2017; Xu et al., 2022; de Bézenac et al., 2019; Xie et al., 2023)), the DDM approach imposes a structural regularization on the translation function realized by GANs. GAN-based

methods sometimes suffer from numerical instability due to its adversarial training nature. More importantly, GANs learn the translation functions but not the continuous intermediate states between the source and target domains, i.e., the *transport trajectories*, yet the latter is critical in many applications such as robot navigation and single-cell evolution inference (Tong et al., 2023; Liu et al., 2023a; 2018).

To overcome these challenges, a natural thought is to use *flow matching* (FM) (Lipman et al., 2023; Albergo et al., 2023) for UDT. FM learns the velocity of the translation function and thus easily recovers the trajectories. In addition, FM methods are friendly to train, using nonlinear least squares instead of min-max adversarial criteria. Nonetheless, using FM in identifiability-driven UDT turns out to be quite nontrivial, as most existing works (e.g., (Zhu et al., 2017; Xu et al., 2022; de Bézenac et al., 2019; Xie et al., 2023; Shrestha & Fu, 2024)) enforce identifiability via imposing constraints/regularization on the translation function—yet, unlike GANs, FM does not have an explicit expression of the function. Shifting such constraints onto the function velocity/trajectory requires completely different designs, which have been elusive in the literature.

**Contributions.** In this work, our interest lies in an FM-based learning framework for DDM-based distribution transport—which we call *diversified flow matching* (DFM). Our detailed contributions are as follows:

*DFM with Transaltion Identifiability.* We custom design the loss function and interpolant (i.e., the function to guide the trajectory of the flow transport) for identifiability-guaranteed DFM. Conventional FM uses nonlinear least squares losses and linear interpolants (Liu et al., 2023b; Tong et al., 2023), which fail to realize DDM as they generate conflicting trajectories among different conditional distribution pairs. We propose to use a nonlinear, private interpolant function for each conditional distribution pair, and show that a *bilevel optimization loss* with this design provably retains the translation identifiability of DDM.

*Tangible Implementation.* We propose an implementation that exploits the non-overlapping property of conditional distributions. This way, we show that the computationally demanding bilevel optimization loss can be recast into a more manageable two-stage approach, consisting of an interpolant learning stage and a flow training stage—both of which admit differentiable unconstrained losses and can be solved using simple back-propagation.

We test our method over synthetic data and real-world applications (i.e., robot crowd route planning and unpaired image translation). The results corroborate with our theoretical analyses and algorithm design.

**Notation.** We largely adhere to established conventions in machine learning; see also Appendix A.

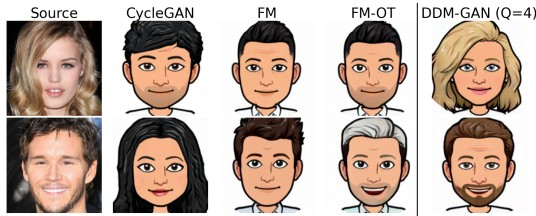

Figure 1: [Columns 2-4] Content misalignment issues in both GAN and FM based UDT (`CycleGAN`(Zhu et al., 2017), `FM`(Lipman et al., 2023), `FM-OT`(Tong et al., 2023)) [Column 5] Result by DDM-GAN (Shrestha & Fu, 2024).

## 2. Background

**Unsupervised Domain Translation.** Consider two data domains (e.g., photos and sketches), denoted by $\mathcal{X} \subseteq \mathbb{R}^d$ and $\mathcal{Y} \subseteq \mathbb{R}^d$, respectively. Assume that there exists a deterministic continuous mapping that translates every $\boldsymbol{x} \in \mathcal{X}$ into its content-aligned counterpart $\boldsymbol{y} \in \mathcal{Y}$, i.e.,

$$\boldsymbol{x} \sim p_{\boldsymbol{x}}, \quad \boldsymbol{y} = \boldsymbol{g}^{\star}(\boldsymbol{x}), \tag{1}$$

where $p_{\boldsymbol{x}}$ is the distribution of $\boldsymbol{x}$ and $\boldsymbol{g}^{\star} : \mathcal{X} \to \mathcal{Y}$ is a differentiable *bijective* map. We stress that there might exist many $\boldsymbol{g}$'s such that $\boldsymbol{g}(\boldsymbol{x}) \in \mathcal{Y}$ for all $\boldsymbol{x} \in \mathcal{X}$, but our interest lies in a content-preserving $\boldsymbol{g}^{\star}$ (e.g., the one that changes the style of handwritten "7" to its printed version but keeps its identity). The goal of UDT is to estimate $\boldsymbol{g}^{\star}$ from the unpaired samples of $p_{\boldsymbol{x}}$ and $p_{\boldsymbol{y}}$.

**Translation via Distribution Transport.** In the literature (Zhu et al., 2017; Park et al., 2020), the UDT problem is generally addressed by finding an invertible translation function $\boldsymbol{g}$ such that the distribution of the translated samples $\boldsymbol{g}(\boldsymbol{x})$ matches that of $\boldsymbol{y}$, i.e.,

$$\text{find } \text{ invertible } \boldsymbol{g} \tag{2}$$
$$\text{subject to} : \boldsymbol{g}_{\#p_{\boldsymbol{x}}} = p_{\boldsymbol{y}},$$

where $\boldsymbol{g}_{\#p_{\boldsymbol{x}}}$ represents the distribution of $\boldsymbol{g}(\boldsymbol{x})$. The criterion can be realized by many computational tools, e.g., GANs (Goodfellow et al., 2014), and more recently, diffusion-based tools such as Shrödinger bridge (De Bortoli et al., 2021), and FM (Lipman et al., 2023). It was noticed in the literature (Galanti et al., 2018; Moriakov et al., 2020) that solving (2) sometimes produces content-misaligned translations—meaning that the desired $\boldsymbol{g}^{\star}$ in the ground-truth translation model (1) is not identified. Fig. 1 (columns 2-4) shows the content misalignment issue that exists in both GANs and FM based UDT.

Many works showed that imposing more structural information on $\boldsymbol{g}$ in (2) could establish content alignment (or, the identifiability of the translation function $\boldsymbol{g}^{\star}$); see, e.g., Benaim & Wolf (2017); Benaim et al. (2018); Xu et al. (2022);

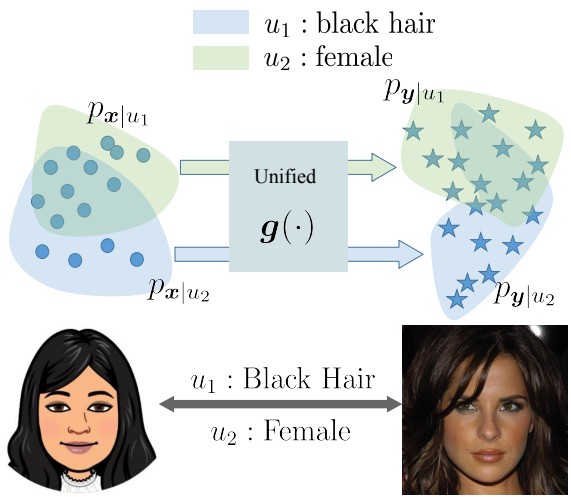

$u_1$ : black hair
$u_2$ : female

$u_1$ : Black Hair
$u_2$ : Female

Figure 2: The idea of DDM. The variable $u^{(q)}$ can often be defined as attributes that are not supposed to change across domains. In (Shrestha & Fu, 2024), it was shown $Q \geq 2$ suffices to underpin the translation identifiability.

Moriakov et al. (2020). Among them, Shrestha & Fu (2024) showed an interesting theoretical result. To elaborate, the method in (Shrestha & Fu, 2024) first defines *corresponding* conditional distributions $p_{\boldsymbol{x}|u=u^{(q)}}$ and $p_{\boldsymbol{y}|u=u^{(q)}}$ for $q \in [Q]$, where $u^{(q)}$ is auxiliary information. For example, for face to cartoon translation in Fig. 2, $u^{(q)}$ could be designed to represent some face attributes, e.g., gender, that are not supposed to change during the translation. Then, they proposed to learn a unified $\boldsymbol{g}$ for matching $Q$ pairs of such conditional distributions using the so-called *diversified distribution matching* (DDM) criterion:

$$\text{(DDM)} \quad \text{find} \quad \text{invertible } \boldsymbol{g} \tag{3}$$
$$\text{subject to } \boldsymbol{g}_{\#p_{\boldsymbol{x}|u^{(q)}}} = p_{\boldsymbol{y}|u^{(q)}}, \forall q \in [Q],$$

where we used $p_{\boldsymbol{x}|u^{(q)}} = p_{\boldsymbol{x}|u=u^{(q)}}$. Shrestha & Fu (2024) introduced the following condition:

**Definition 2.1** (Sufficiently Diverse Condition (SDC)). For any two disjoint sets $\mathcal{A}, \mathcal{B} \subset \mathcal{X}$, where $\mathcal{A}$ and $\mathcal{B}$ are connected, open, and non-empty, there exists a $u_{(\mathcal{A},\mathcal{B})} \in \{u_1, \ldots, u^{(q)}\}$ such that $\int_{\mathcal{A}} p_{\boldsymbol{x}|u_{(\mathcal{A},\mathcal{B})}}(\boldsymbol{x})d\boldsymbol{x} \neq \int_{\mathcal{B}} p_{\boldsymbol{x}|u_{(\mathcal{A},\mathcal{B})}}(\boldsymbol{x})d\boldsymbol{x}$. Then, the set of conditional distributions $\{p_{\boldsymbol{x}|u^{(q)}}\}_{q=1}^{Q}$ is called *sufficiently diverse*.

Under the SDC, there are at least two PDFs among $\{p_{\boldsymbol{x}|u^{(q)}}\}_{q=1}^{Q}$ that are sufficiently different over any $\mathcal{A}$ and $\mathcal{B}$. It was also shown that

**Theorem 2.2** (Translation Identifiability). *(Shrestha & Fu, 2024) Suppose that $\{p_{\boldsymbol{x}|u^{(q)}}\}_{q=1}^{Q}$ satisfies the SDC. Let $\widehat{\boldsymbol{g}}$ be any optimal solution of the DDM criterion* (3). *Then, we have $\widehat{\boldsymbol{g}} = \boldsymbol{g}^{\star}$, a.e.*

In addition, it was also shown that the DDM criterion is robust to small violations of the SDC (see Appendix B.1).

**DDM-GAN and Challenges.** In (Shrestha & Fu, 2024), Problem (3) was solved using a GAN-based framework:

$$\min_{\boldsymbol{g}} \max_{\{\boldsymbol{d}^{(q)}\}} \sum_{q=1}^{Q} \Pr(u^{(q)}) \Big( \mathbb{E}_{\boldsymbol{y} \sim p_{\boldsymbol{y}|u^{(q)}}} \Big[ \log \boldsymbol{d}^{(q)}(\boldsymbol{y}) \Big]$$
$$+ \mathbb{E}_{\boldsymbol{x} \sim p_{\boldsymbol{x}|u^{(q)}}} \Big[ \log \Big( 1 - \boldsymbol{d}^{(q)}(\boldsymbol{g}(\boldsymbol{x})) \Big) \Big] \Big), \tag{4}$$

where $\boldsymbol{d}^{(q)}$ is a discriminator for the $q$th pair of conditional distributions. Cycle-consistency and backward translation were also used to enforce invertibility of $\boldsymbol{g}$, which is omitted here for conciseness. The DDM-GAN formulation showed promising results (see Fig. 1), but two main challenges exist: First, GAN-based training is sometimes numerically unstable due to the min-max nature. Second, more importantly, the learned deterministic $\boldsymbol{g}$ does not contain the trajectory reflecting how $p_{\boldsymbol{x}}$ was changed to $p_{\boldsymbol{y}}$, but such trajectories are critical information for a number of domain translation problems, e.g., robot navigation and single-cell evolution inference (Tong et al., 2023; Liu et al., 2023a; 2018).

## 3. Proposed Approach

As mentioned, besides GANs, diffusion and flow based methods can also be used for distribution transport. The latter genre is known for their relatively simple training processes and the ability to reveal the transport trajectories. Hence, we are motivated to design an FM (Lipman et al., 2023) based UDT method for carrying out the DDM principle (3). This turns out to be a nontrivial task. To see this, we start with some preliminaries of FM.

### 3.1. Preliminaries on Flow Matching

FM is an instance of a class of generative models called *continuous normalizing flow* (CNF) (Lipman et al., 2023). CNFs learn a time-varying differentiable map $\boldsymbol{f}_t : \mathbb{R}^d \to \mathbb{R}^d, \forall t \in [0, 1]$, called the *flow*, such that $\boldsymbol{f}_t(\boldsymbol{x}) = \boldsymbol{z}_t$, where $\boldsymbol{z}_0 = \boldsymbol{x}, \boldsymbol{z}_1 = \boldsymbol{y}$. The flow $\boldsymbol{f}_t$ has a velocity field $\boldsymbol{v}_t : \mathbb{R}^d \to \mathbb{R}^d$:

$$\boldsymbol{v}_t(\boldsymbol{f}_t(\boldsymbol{x})) = \frac{d}{dt} \boldsymbol{f}_t(\boldsymbol{x}). \tag{5}$$

Using the vector field $\boldsymbol{v}_t$, one can easily translate a given sample $\boldsymbol{x}$ to any intermediate state in the trajectory from $\boldsymbol{x}$ to its corresponding $\boldsymbol{y}$:

$$\boldsymbol{f}_t(\boldsymbol{x}) = \boldsymbol{x} + \int_0^t \boldsymbol{v}_s(\boldsymbol{z}_s)ds, \ t \in [0, 1] \tag{6}$$

and we have $\boldsymbol{f}_1(\boldsymbol{x}) = \boldsymbol{g}(\boldsymbol{x}) = \boldsymbol{y}$. In practice, $\boldsymbol{v}_t$ for transporting $p_{\boldsymbol{x}}$ to $p_{\boldsymbol{y}}$ is parameterized by a neural network and

learned via the following nonlinear least squares loss (Albergo et al., 2023; Lipman et al., 2023; Liu et al., 2023b):

$$\min_{\boldsymbol{v}_t} \underbrace{\mathbb{E}_{\substack{\boldsymbol{x},\boldsymbol{y}\sim\rho(\boldsymbol{x},\boldsymbol{y}),\\ t\sim\mathrm{Unif}([0,1])\\ \overline{\boldsymbol{z}}_t=\boldsymbol{I}(\boldsymbol{x},\boldsymbol{y},t)}} \|\boldsymbol{v}_t(\overline{\boldsymbol{z}}_t) - \partial_t\boldsymbol{I}(\boldsymbol{x},\boldsymbol{y},t)\|_2^2}_{\mathcal{L}_{\mathrm{FM}}(\boldsymbol{v}_t,\boldsymbol{I},\rho)}, \quad (7)$$

where $\rho(\boldsymbol{x},\boldsymbol{y})$ is any joint distribution of $\boldsymbol{x},\boldsymbol{y}$ such that $\int_{\mathcal{X}}\rho(\boldsymbol{x},\boldsymbol{y})d\boldsymbol{x} = p_{\boldsymbol{y}}(\boldsymbol{y})$ and $\int_{\mathcal{Y}}\rho(\boldsymbol{x},\boldsymbol{y})d\boldsymbol{y} = p_{\boldsymbol{x}}(\boldsymbol{x})$, and $\boldsymbol{I} : \mathbb{R}^d \times \mathbb{R}^d \times [0,1] \to \mathbb{R}^d$ is the so-called *interpolant function* which is differentiable with respect to $t$ and satisfies $\boldsymbol{I}(\boldsymbol{x},\boldsymbol{y},0) = \boldsymbol{x}$ and $\boldsymbol{I}(\boldsymbol{x},\boldsymbol{y},1) = \boldsymbol{y}$. The independent coupling $\rho(\boldsymbol{x},\boldsymbol{y}) = p_{\boldsymbol{x}}(\boldsymbol{x})p_{\boldsymbol{y}}(\boldsymbol{y})$ and the linear interpolant

$$\boldsymbol{I}^{\mathrm{linear}}(\boldsymbol{x},\boldsymbol{y},t) = (1-t)\boldsymbol{x} + t\boldsymbol{y} \quad (8)$$

are most commonly used in the FM literature.

### 3.2. Challenges of FM-based DDM

While FM appears to circumvent some well-known challenges of GANs—such as the instability of min-max optimization—it introduces its own distinct difficulties when used to realize the DDM criterion in (3). To explain, let us start with the following definitions:

**Definition 3.1** (Transport of Measures). A vector field $\boldsymbol{v}_t$ is said to transport a distribution $\omega$ to $\eta$ if the corresponding flow $\boldsymbol{g}_{\boldsymbol{v}} = \boldsymbol{f}_1$ (cf. Eq. (6)) satisfies: $[\boldsymbol{g}_{\boldsymbol{v}}]_{\#\omega} = \eta$.

**Definition 3.2** (DDM Satisfaction). A vector field $\boldsymbol{v}_t$ is said to satisfy DDM in (3) if it transports $p_{\boldsymbol{x}|u^{(q)}}$ to $p_{\boldsymbol{y}|u^{(q)}}$ for all $q \in [Q]$, i.e., $[\boldsymbol{g}_{\boldsymbol{v}}]_{\#p_{\boldsymbol{x}|u^{(q)}}} = p_{\boldsymbol{y}|u^{(q)}}, \forall q \in [Q]$.

Next, we show that the classical interpolant in (8) fails to attain DDM satisfaction.

**Classic Linear Interpolant Fails.** A naive way to implement DDM using FM is as follows:

$$\underset{\boldsymbol{v}_t}{\mathrm{minimize}} \sum_{q=1}^{Q} \mathcal{L}_{\mathrm{FM}}(\boldsymbol{v}_t, \boldsymbol{I}^{\mathrm{linear}}, \rho_q), \quad (9)$$

where $\rho_q := \rho(\boldsymbol{x},\boldsymbol{y}|u^{(q)}) = p_{\boldsymbol{x}|u^{(q)}}(\boldsymbol{x})p_{\boldsymbol{y}|u^{(q)}}(\boldsymbol{y})$. The goal of (9) is to enforce a *unified* vector field to transport $p_{\boldsymbol{x}|u^{(q)}}$ to $p_{\boldsymbol{y}|u^{(q)}}$ for each $u^{(q)}$. Unfortunately, the loss (9) cannot attain this goal in general. Fig. 3 shows a typical failure case. The source and target distributions are both Gaussian mixtures with two modes. There, the red "x" are supposed to be translated to red "●" (same for the blue modes). However, the linear interpolants for two pairs of conditional distributions intersect at $t = 1/2$. This intersection results in $\widehat{\boldsymbol{v}}_{\frac{1}{2}}(\frac{1}{2}\boldsymbol{x} + \frac{1}{2}\boldsymbol{y}) = 2(\frac{1}{2}\boldsymbol{x} + \frac{1}{2}\boldsymbol{y} - \mathbb{E}[\boldsymbol{x}])$, where $\widehat{\boldsymbol{v}}_t$ is the optimal solution of (9) (see Fig. 3(d) and Appendix 3.4 for derivation; also see similar visualizations in (Liu

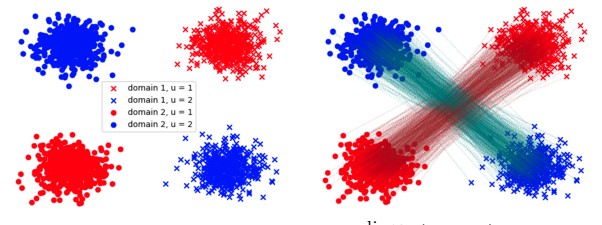

(a) Left: $p_{\boldsymbol{x}|u^{(q)}}$. Right:(b) $\boldsymbol{I}^{\mathrm{linear}}(\boldsymbol{x},\boldsymbol{y},t)$ as interpolant trajectories.
$p_{\boldsymbol{y}|u^{(q)}}$.

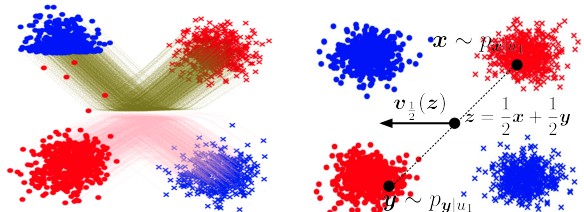

(c) Actual trajectories computed from (9). (d) $\widehat{\boldsymbol{v}}_{1/2}(0.5\boldsymbol{x} + 0.5\boldsymbol{y})$.

Figure 3: (a) Samples of two pairs of conditional distributions. (b) Linear interpolant at $t \in [0,1]$ for interpolant trajectories. (c) Actural trajectories learned by solving (9). (d) $\widehat{\boldsymbol{v}}_{\frac{1}{2}}$ points towards the $-\mathbb{E}[\boldsymbol{x}]$.

et al., 2023b)). This implies that the samples from $p_{\boldsymbol{x}|u_1}$ gets "reflected" back to $p_{\boldsymbol{y}|u_2}$ and that from $p_{\boldsymbol{x}|u_2}$ to $p_{\boldsymbol{y}|u_1}$, i.e., "x" to "●" and "x" to "●" as shown in Fig. 3(c).

**Using Private and Learnable Interpolants.** The previous example shows that the commonly used interpolant $\boldsymbol{I}^{\mathrm{linear}}(\boldsymbol{x},\boldsymbol{y},t) = (1-t)\boldsymbol{x} + t\boldsymbol{y}$ does not work for DDM. Note that the interpolant "guides" the velocity, and it is well known that the velocity is the conditional mean of the time-derivative of the interpolant (Albergo et al., 2023). Hence, to learn a legitimate $\boldsymbol{v}_t$ that transports $p_{\boldsymbol{x}|u^{(q)}}$ to $p_{\boldsymbol{y}|u^{(q)}}$ for all $q$ that satisfies the DDM criterion (3), a suitable interpolant needs to be selected.

To this end, we propose the following strategy: First, we let each pair $p_{\boldsymbol{x}|u^{(q)}}$ and $p_{\boldsymbol{y}|u^{(q)}}$ use their own *private* interpolant, denoted by $\boldsymbol{I}^{(q)}$. Second, we design the $\boldsymbol{I}^{(q)}$'s to be nonlinear, learnable interpolants. To proceed, we define a set of learnable $\mathcal{I}$ as follows:

$$\mathcal{I} = \{\boldsymbol{I}_{\boldsymbol{\theta}}(\boldsymbol{x},\boldsymbol{y},t) : \mathbb{R}^d \times \mathbb{R}^d \times [0,1] \to \mathbb{R}^d \mid \boldsymbol{I}_{\boldsymbol{\theta}}(\boldsymbol{x},\boldsymbol{y},0) = \boldsymbol{x},$$
$$\boldsymbol{I}_{\boldsymbol{\theta}}(\boldsymbol{x},\boldsymbol{y},1) = \boldsymbol{y}, \boldsymbol{I}_{\boldsymbol{\theta}} \text{ differentiable w.r.t. } t\}$$

Using the private interpolant, a natural formulation to realize the DDM (3) appears to be the following:

$$\underset{\boldsymbol{v}_t,\{\boldsymbol{I}^{(q)}\}_{q=1}^{Q}}{\mathrm{minimize}} \sum_{q=1}^{Q} \underbrace{\mathcal{L}_{\mathrm{FM}}\left(\boldsymbol{v}_t, \boldsymbol{I}^{(q)}, \rho(\boldsymbol{x},\boldsymbol{y}|u^{(q)})\right)}_{\mathcal{L}_{\mathrm{FM}}^{(q)}}, \quad (10)$$

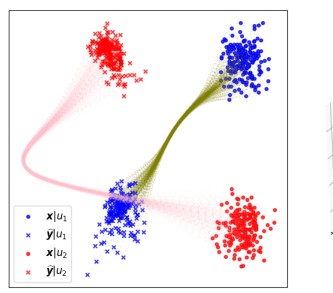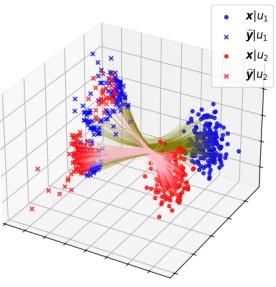

Figure 4: [Left] Success case of solving (10). [Right] Failure case of solving (10).

where $\boldsymbol{I}^{(q)} \in \mathcal{I}$ and has learnable parameters $\boldsymbol{\theta}_q$. The hope is that $\boldsymbol{I}^{(q)}$ will be learned in a way such that $\boldsymbol{v}_t$ simultaneously minimizes each $\mathcal{L}_{\mathrm{FM}}^{(q)}$ for $q = 1, \ldots, Q$ in (10) in order to attain DDM satisification.

**Pitfall of Loss** (10). However, it turns out that (10) is *not* a correct criterion. Fig. 4 shows the result of solving (10) on the Gaussian mixture example (see details in Appendix B.4). One can see that it sometimes works [Left] yet sometimes fails [Right]—no matter how hard one tunes the optimization algorithm for solving (10). This drives us to discover the following fact:

**Fact 3.3.** The problem formulation in (10) is *not* equivalent to the DDM criterion in (3).

To explain, note that we hope using the loss in (10) to find

$$\widehat{\boldsymbol{v}}_t(\boldsymbol{z}) = \mathbb{E}[\partial_t \widehat{\boldsymbol{I}}^{(q)}(\boldsymbol{x}, \boldsymbol{y}, t) \mid \widehat{\boldsymbol{I}}^{(q)}(\boldsymbol{x}, \boldsymbol{y}, t) = \boldsymbol{z}], \quad (11)$$

for all $q \in [Q]$, where the expectation is taken over $\rho(\boldsymbol{x}, \boldsymbol{y}|u^{(q)})$. Under the model in (1), such $\widehat{\boldsymbol{v}}_t$ and $\widehat{\boldsymbol{I}}^{(q)}$ for all $q$ could exist (which we will discuss later in more details). However, when minimizing the loss(10), it is possible that one finds $(\widetilde{\boldsymbol{v}}_t, \widetilde{\boldsymbol{I}}^{(q)})$ for certain $q$'s such that $\mathcal{L}_{\mathrm{FM}}^{(q)}(\widetilde{\boldsymbol{v}}_t, \widetilde{\boldsymbol{I}}^{(q)}) \ll \mathcal{L}_{\mathrm{FM}}^{(q)}(\widehat{\boldsymbol{v}}_t, \widehat{\boldsymbol{I}}^{(q)})$ yet $\widetilde{\boldsymbol{v}}_t \neq \mathbb{E}[\partial_t \widetilde{\boldsymbol{I}}^{(q)}(\boldsymbol{x}, \boldsymbol{y}, t) \mid \widetilde{\boldsymbol{I}}^{(q)}(\boldsymbol{x}, \boldsymbol{y}, t) = \boldsymbol{z}]$. This pathological case could happen because $\boldsymbol{I}^{(q)}$ is a learnable term—whose scale can be changed to attain an overall small $\mathcal{L}_{\mathrm{FM}}^{(q)}$. However, such small loss values do not reflect the real goal of distribution transport. In other words, due to the changeable $\boldsymbol{I}^{(q)}$ and the setting $Q > 1$, having a small value of $\mathcal{L}_{\mathrm{FM}}^{(q)}(\widetilde{\boldsymbol{v}}_t, \widetilde{\boldsymbol{I}}^{(q)})$ does not mean (11) is met for $q$, which makes (10) problematic for enforcing DDM.

### 3.3. Proposed Criterion: A Bilevel Learning Loss

The above discussion shows that we need (11) to hold for each $q$. Hence, instead of using a sum of LS-type loss, we propose the following *diversified flow matching* (DFM)

criterion:

$$\underset{\boldsymbol{v}_t, \boldsymbol{v}_t^{(q)}, \boldsymbol{I}^{(q)}}{\text{minimize}} \sum_{q=1}^{Q} \|\boldsymbol{v}_t^{(q)} - \boldsymbol{v}_t\|^2 + \mathbf{1}_{\mathcal{I}}[\boldsymbol{I}^{(q)}] \quad (12a)$$

$$\text{subject to}: \boldsymbol{v}_t^{(q)} = \arg\min_{\boldsymbol{w}_t^{(q)}} \mathcal{L}_{\mathrm{FM}}\left(\boldsymbol{w}_t^{(q)}, \boldsymbol{I}^{(q)}, \rho^{(q)}\right), \quad (12b)$$

where $\rho^{(q)} = \rho(\boldsymbol{x}, \boldsymbol{y}|u^{(q)})$ for short and $\mathbf{1}_{\mathcal{I}}[\boldsymbol{I}]$ is the indicator function of set $\mathcal{I}$. Problem (12) is a bilevel optimization problem, where we ensure DDM satisfaction by using two constraints (12b). The *lower level optimization* (12b) introduces private vector field $\boldsymbol{v}_t^{(q)}$ for each $q$ and the *upper level optimization* forces consensus among all $\boldsymbol{v}_t^{(q)}$. Consequently, (11) holds for all $q \in [Q]$ when (12) is optimally solved—recovering the DDM criterion in (3). This leads to the following proposition:

**Proposition 3.4.** *Suppose that there exists a flow $\boldsymbol{f}_t^\star :$ $\mathbb{R}^d \to \mathbb{R}^d$, continuously differentiable in time and space, from $p_{\boldsymbol{x}}$ to $p_{\boldsymbol{y}}$ such that $\boldsymbol{f}_1^\star = \boldsymbol{g}^\star$. Suppose there exist a diffeomorphism from the standard Gaussian $\mathcal{N}(0, I_d)$ to $p_{\boldsymbol{x}|u^{(q)}}, \forall q$. Let $\widehat{\boldsymbol{v}}_t$ denote a solution of Problem (12) and denote $\boldsymbol{g}_{\widehat{\boldsymbol{v}}}(\boldsymbol{x}) = \boldsymbol{x} + \int_{t=0}^{t=1} \widehat{\boldsymbol{v}}_t(\boldsymbol{z}_t)dt$. Then, under the same model in (1), when $\{p_{\boldsymbol{x}|u^{(q)}}\}_{q=1}^{Q}$ satisfies the SDC, we have $\boldsymbol{g}_{\widehat{\boldsymbol{v}}} = \boldsymbol{g}^\star$, holds a.e.*

Proposition 3.4 shows that it is viable to use an FM-based loss to attain the same conclusion of Theorem 2.2. Note that Theorem 2.2 was established under the premise that DDM is attained, yet Proposition 3.4 specifically needs the distribution matching part to be realized by FM. This gap is filled by the bilevel loss design in (12).

### 3.4. Implementation: Exploiting Structural Constraint

Unlike (10), which suffers from theoretical flaws, (12) is theoretically sound in establishing translation identifiability. However, it requires solving a bilevel optimization problem. While this can be tackled using off-the-shelf techniques such as implicit gradients and gradient unrolling (see (Zhang et al., 2024)), computational efficiency remains a concern.. In addition, for problems where the lower level optimization part is nonconvex and intractable, convergence of bilevel optimization is hard to guarantee.

To avoid these computational barriers, we propose to simplify the Problem (12) by exploiting the structural property of conditional distributions that naturally arises in many cases. Specifically, the auxiliary information $u$ often correspond to semantic attributes/labels (e.g., gender for image translation), which induce roughly non-overlapping clusters. To utilize this structure, let us assume the following:

**Assumption 3.5** (Non-overlapping Supports). For $\{p_{\boldsymbol{x}|u^{(q)}}\}_{q=1}^{Q}$, we have $\mathrm{supp}(p_{\boldsymbol{x}|u_i}) \cap \mathrm{supp}(p_{\boldsymbol{x}|u_j}) = \phi$

and $\mathrm{supp}(p_{\boldsymbol{y}|u_i}) \cap \mathrm{supp}(p_{\boldsymbol{y}|u_j}) = \phi, \forall i \neq j$, where $\mathrm{supp}(p) = \{\boldsymbol{z} \mid p(\boldsymbol{z}) > 0\}$.

Under the above assumption, it is possible to design probability paths $p_{u^{(q)}}^t, \forall q$, that satisfy the boundary conditions $p_{u^{(q)}}^0 = p_{\boldsymbol{x}|u^{(q)}}$ and $p_{u^{(q)}}^1 = p_{\boldsymbol{y}|u^{(q)}}$, such that $p_{u^{(q)}}^t$'s supports are non-overlapping among all $q \in [Q]$. Learning unified vector field that simultaneously follow the $p_{u^{(q)}}^t$ for all $q \in [Q]$ will then ensure that the vector field satisfies DDM.

Designing such non-intersecting $p_{u^{(q)}}^t$ can be achieved by designing the interpolants themselves. This is due to the well-known fact:

**Fact 3.6.** (Albergo et al., 2023) The distribution $p_{u^{(q)}}^t$ is the same as the probability distribution of the random variable $\boldsymbol{I}^{(q)}(\boldsymbol{x}, \boldsymbol{y}, t)$ where $\boldsymbol{x}, \boldsymbol{y} \sim \rho(\boldsymbol{x}, \boldsymbol{y}|u^{(q)})$.

Therefore, it suffices to select interpolant $\boldsymbol{I}^{(q)}$ for all $q$ that do not "intersect" with each other in the following sense:

**Definition 3.7** (Non-intersecting Interpolants). The interpolants $\{\boldsymbol{I}^{(q)}\}_{q=1}^Q$ are *non-intersecting* if $\forall i, j \in [Q], i \neq j$,

$$\boldsymbol{I}^{(i)}(\boldsymbol{x}^{(i)}, \boldsymbol{y}^{(i)}, t) \neq \boldsymbol{I}^{(j)}(\boldsymbol{x}^{(j)}, \boldsymbol{y}^{(j)}, t),$$

for all $(\boldsymbol{x}^{(q)}, \boldsymbol{y}^{(q)}) \in \mathrm{supp}(\rho(\boldsymbol{x}, \boldsymbol{y}|u^{(q)})), q \in \{i, j\}$.

As we will see, non-intersecting interpolants will allow us to greatly simplify the bilevel loss so that the unified vector field can be learnt efficiently.

**Learning Non-intersecting Interpolants.** A major benefit of exploiting the non-overlapping support structure in Assumption 3.5 as follows: Under Assumption 3.5, assume that $\boldsymbol{I}^{(q)}$ for all $q \in [Q]$ are learnable universal path representers. Then, there exists a set of $\check{\boldsymbol{I}}^{(q)}$ for $q \in [Q]$ that are non-intersecting. This is simply because the starting and ending points mapped by $\check{\boldsymbol{I}}^{(i)}$ and $\check{\boldsymbol{I}}^{(j)}$ are completely different. Further, using Assumption 3.5, one can use a unified $\boldsymbol{I} = \boldsymbol{I}^{(q)}$ to express the non-intersecting interpolants for transporting $p_{\boldsymbol{x}|u^{(q)}}$ to $p_{\boldsymbol{y}|u^{(q)}}$ for all $q \in [Q]$; that is,

$$\check{\boldsymbol{I}}(\boldsymbol{x}, \boldsymbol{y}, t) = \sum_{q=1}^Q \mathbf{1}_{\boldsymbol{x}, \boldsymbol{y} \in \mathrm{supp}(\rho^{(q)})} \check{\boldsymbol{I}}^{(q)}(\boldsymbol{x}, \boldsymbol{y}, t). \quad (13)$$

To find a non-intersecting unified interpolant, we use the following learning criterion:

$$\check{\boldsymbol{I}} = \arg\min_{\boldsymbol{I}} \sum_{i=1}^{Q-1} \sum_{j=i+1}^Q \mathbb{E}_{t_1, t_2}[\gamma_{\sigma_2}(|t_1 - t_2|) \times \quad (14)$$

$$\gamma_{\sigma_1}(\|\boldsymbol{I}(\boldsymbol{x}^{(i)}, \boldsymbol{y}^{(i)}, t_1) - \boldsymbol{I}(\boldsymbol{x}^{(j)}, \boldsymbol{y}^{(j)}, t_2)\|_2)],$$

where $\gamma_\sigma(a) = \max(\exp(-a^2/2\sigma^2), \eta)$ and $\eta > 0$ is a small constant. Simply speaking, Problem (14) tries to push

the values of $\boldsymbol{I}(\boldsymbol{x}^{(i)}, \boldsymbol{y}^{(i)}, t_1)$ and $\boldsymbol{I}(\boldsymbol{x}^{(j)}, \boldsymbol{y}^{(j)}, t_2)$ apart, if the two values are close in time and space.

**Simplifying the Bilevel Loss.** Let $\check{\boldsymbol{I}}$ denote the learned non-intersecting interpolant. Then (12) can be reformulated as follows:

$$\min_{\boldsymbol{v}_t, \boldsymbol{v}_t^{(q)}} \sum_{q=1}^Q \|\boldsymbol{v}_t^{(q)} - \boldsymbol{v}_t\|^2 \quad (15a)$$

$$\text{subject to } \boldsymbol{v}_t^{(q)} = \arg\min_{\boldsymbol{w}_t} \mathcal{L}_{\mathrm{FM}}\left(\boldsymbol{w}_t, \check{\boldsymbol{I}}, \rho^{(q)}\right) \quad (15b)$$

In contrast to (12), $\check{\boldsymbol{I}}$ is not learnable in Problem (15). This allows us to further re-express the formulation as:

$$\min_{\boldsymbol{v}_t} \sum_{q=1}^Q \mathcal{L}_{\mathrm{FM}}(\boldsymbol{v}_t, \check{\boldsymbol{I}}, \rho(\boldsymbol{x}, \boldsymbol{y}|u^{(q)})), \quad (16)$$

Note that we have eliminated the slack variable representing the private $\boldsymbol{v}^{(q)}$ and the consensus loss in (15a), as no private $\boldsymbol{I}^{(q)}$ is needed in our loss function.

The proposed algorithm is referred to as DFM and is detailed in Algorithm 1 in Appendix B.

## 4. Related Works

**UDT via Distribution Transport.** Distribution transport is arguably the most widely used approach in UDT (as well as closely related techniques such as domain adaptation (Long et al., 2016)). In the literature, distribution transport is realized using various methods such as minimization of maximum mean discrepancy (MMD) (Long et al., 2016), GANs (Zhu et al., 2017; Huang et al., 2018; Choi et al., 2020), bridge matching (De Bortoli et al., 2021; Liu et al., 2023a; De Bortoli et al., 2024), and FM (Liu et al., 2023b; Eyring et al., 2024).

**Translation Identifiability.** The non-uniqueness of the distribution transport maps in UDT is a well-known issue (Moriakov et al., 2020; Shrestha & Fu, 2024; Galanti et al., 2018). Many hypothesized that the desired map $\boldsymbol{g}^\star$ is likely to be the *optimal transport* (OT); see (Liu et al., 2023b) and (De Bortoli et al., 2021; 2024). While OT maps have been shown to be effective in some applications such as single cell analysis (Bunne et al., 2024), it is in general not clear if $\boldsymbol{g}^\star$ is the OT map. There also exist many empirical ways to constrain the feasible set of the transport maps, making the found $\hat{\boldsymbol{g}}$ with intended behaviors (Amodio & Krishnaswamy, 2019; Xu et al., 2022). Recently, (Shrestha & Fu, 2024) showed that it is possible to identify a general (non-OT) translation function under the SDC.

**Bridge and Flow Matching for UDT.** Following the rise of diffusion and FM based generative models, many works have emerged to apply the continuous density transport

perspective onto UDT. Two major classes are notable. The first class is the FM-based approaches that use deterministic continuous time process characterized by an ODE (Liu et al., 2023b; Eyring et al., 2024; Kapuśniak et al., 2024; Kornilov et al., 2024; Gazdieva et al., 2023). The second is diffusion and schrödinger bridges (Liu et al., 2023a; De Bortoli et al., 2024; Sasaki et al., 2021), characterized by SDEs. These methods do not consider the translation identifiability and thus issues in Fig. 1 arise. In addition, they are designed to transport among only one pair of distributions, which is hard to use for DDM as in this work.

**Using Auxiliary Information in FM.** Auxiliary information has been incorporated into FM in the context of learning conditional generative models (Atanackovic et al., 2024; Zhu & Lin, 2024). These approaches typically learn separate vector fields $v_t(\cdot \mid u^{(q)})$ for each conditioning variable $u^{(q)}$, treating the auxiliary information as a condition. As a result, they avoid the need to construct a unified $v_t$ across all conditional pairs and do not require designing nonlinear, learnable interpolants. This makes them fundamentally different from the setting considered in this work.

# 5. Experiments

**Interpolant Construction.** In order to construct the learnable $\mathcal{I}$, we parametrize $I_\theta$ using the following form:

$$I_\theta(x, y, t) = (1 - t)x + ty + t(1 - t)\gamma_\theta(x, y, t), \quad (17)$$

where $\gamma_\theta : \mathbb{R}^d \times \mathbb{R}^d \times [0, 1] \to \mathbb{R}^d$ is a neural network. Note that $I_\theta$ constructed this way satisfies the boundary conditions by design, i.e., $I_\theta(x, y, 0) = x$ and $I_\theta(x, y, 1) = y$. Similar constructions have been used in the literature (Kapuśniak et al., 2024).

**Baselines.** The major baselines used throughout this section are plain-vanilla flow matching (FM) (Albergo et al., 2023; Liu et al., 2023b; Lipman et al., 2023) and FM with minim-batch optimal transport coupling (FM-OT) (Pooladian et al., 2023; Tong et al., 2023). We also modify the plain-vanilla FM and FM-OT to incorporate auxiliary variables following Eq. (9)—to show that without custom designed interpolants, simply using auxiliary variables does not attain DDM satisfaction or translation identifiability. The two modified algorithms are referred to as FM-cond and FM-cond-OT, respectively. For Sec. 5.3 and the navigation experiment there, we further use metric flow matching (MFM-OT), and its auxiliary-variable modified version MFM-cond-OT (also following Eq. (9)). For Sec. 5.2 and the image translation task there, we also include GAN-based CycleGAN (Zhu et al., 2017) and DDM-GAN (Shrestha & Fu, 2024) and diffusion-based baselines SDEdit (Meng et al., 2021) and EGSDE (Zhao et al., 2022).

## 5.1. Synthetic Data Validation

**Setting.** We use two layer MLP with 64 hidden units and SeLU activations to represent $v_t(\cdot; \phi)$ as well as $I_\theta$. We use an Adam optimizer with an initial learning rate of 0.001 for $v_t$ and 0.0001 for $I_\theta$. We use a batch size of 512. Details of the proposed algorithm is presented in Algorithm 1. We run both phases of Algorithm 1 for 2000 iterations.

**Metrics.** We use the Earth Mover's Distance (EMD) to assess the distribution matching between $p_{y|u^{(q)}}$ and $p_{\widehat{y}|u^{(q)}}, \forall q$. Similarly, we use translation error to assess the identifiability with respect to the true translation function $g^\star$, which is defined as follows

$$\text{Translation Error(TE)} = \frac{1}{N} \sum_{n=1}^{N} \|\widehat{y}_n - y_n\|_2$$

where $y_n = g^\star(x_n)$ target translation and $\widehat{y}_n = g_{\widehat{v}}(x_n) = x_n + \int_{t=0}^{1} \widehat{v}_t(z_t)dt$ is the predicted translation.

**3D Gaussian blobs.** Fig. 5 shows the results of a setting where we generate Gaussian mixtures $p(x)$ with $d = 3$ and use $y = g^\star(x) = -x$ to generate samples from $p(y)$, where each Gaussian component is with unit variance. Trajectories obtained by the baselines FM-cond and FM-cond-OT show the "reflection" effect discussed in Fig. 3, which results in $p(\widehat{y}|u^{(q)})$ being very different from $p(y|u^{(q)})$. However, the proposed method successfully finds vector fields that circumvent the reflection issue associated with this setting, correctly transporting $p_{x|u^{(q)}}$ to $p_{y|u^{(q)}}, \forall q \in \{1, 2\}$.

**2D Gaussian blobs.** Fig. 6 shows the performance of DFM and baselines when $d = 2$. Note that the $d = 2$ case is arguably more challenging than the $d = 3$ case, as the transport trajectories have less space to explore for collision avoidance. Nonetheless, one can see that DFM learns a $v_t$ such that the two Gaussian blobs travel at *different* speed (see the color bar of $t$ in Fig. 6) to avoid collision and reflection, successfully transporting $p_{x|u^{(q)}}$ to $p_{y|u^{(q)}}, \forall q \in \{1, 2\}$. This interesting behavior is not acquired by FM-cond or FM-cond-OT, showing the importance of designing $\mathcal{I}$.

**Quantitative Results.** Table 1 shows the mean and standard deviations of EMD and TE attained by various methods averaged over 10 trials. It shows that the proposed method successfully transports $p_{x|u^{(q)}}$ to $p_{y|u^{(q)}}, \forall q$ and identifies $g^\star$ more accurately relative to the baselines.

## 5.2. Image Translation

In this subsection, we demonstrate the efficacy of the proposed method on an important UDT task, namely, unpaired image to image translation. We use images of human faces from the CelebAHQ dataset (Karras et al., 2017) with

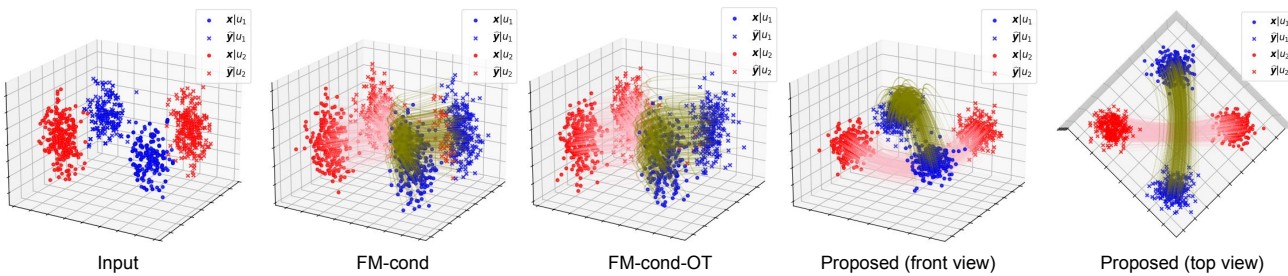

Figure 5: Trajectories returned by all methods for the 3D synthetic data.

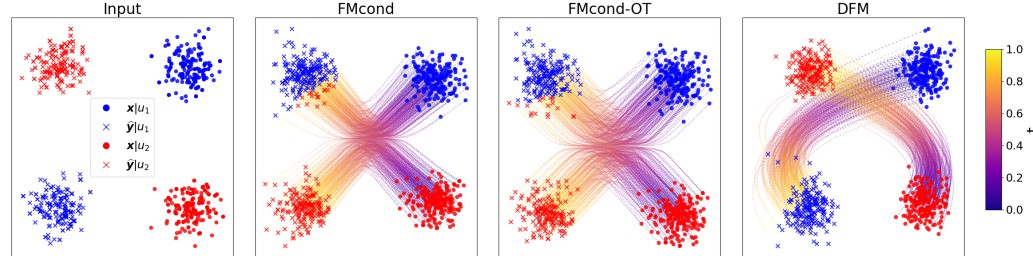

Figure 6: Trajectories returned by all methods for the 2D synthetic data. Colorbar indicates time $t$.

Table 1: `EMD` and `TE` attained by the proposed method and baselines for the synthetic data.

| Method | EMD | | TE | |
|---|---|---|---|---|
| | 2D Gaussians | 3D Gaussians | 2D Gaussians | 3D Gaussians |
| FM-cond | $9.64 \pm 0.51$ | $9.59 \pm 0.45$ | $9.95 \pm 0.43$ | $10.115 \pm 0.388$ |
| FM-cond-OT | $9.56 \pm 0.41$ | $9.44 \pm 0.49$ | $9.86 \pm 0.37$ | $9.994 \pm 0.428$ |
| DFM (proposed) | $\mathbf{0.31 \pm 0.06}$ | $\mathbf{0.54 \pm 0.05}$ | $\mathbf{2.42 \pm 0.04}$ | $\mathbf{3.196 \pm 0.034}$ |

$30,000$ images as the source data $p_{\boldsymbol{x}}$ and Bitmoji faces with $4084$ images (Mozafari, 2020) as the target data $p_{\boldsymbol{y}}$. To avoid data imbalance, we only use randomly selected $5000$ images from CelebAHQ. Both image domains are resized to have a size of $256 \times 256$. The Bitmoji faces are first center-cropped to $80\%$ of the original image before resizing. All FM-based methods are trained on the latent space of the VAE from Stable Diffusion v1 (Rombach et al., 2022). We use FID to measure the distribution matching performance, while translation identifiability is visually checked from content alignment. The auxiliary information used for this task is the gender, i.e., $u_1 = $ "male" and $u_2 = $ "female". More details and hyperparameter settings are in Appendix D.1.

**DDM-GAN's Convergence Issues.** As discussed in Sec. 2, GANs could encounter convergence challenges in some scenarios. Fig. 7 shows a case where the dataset size is up to 5000 and $Q = 2$. One can see that FID exhibits quite erratic behaviors when the number of iterations increases. Note that FID $\geq 70$ is unacceptable quality (see Appendix B for an example). As we mentioned, such hardness in adversarial optimization is a reason motivating our DFM approach. Therefore, we present the best FID attained by

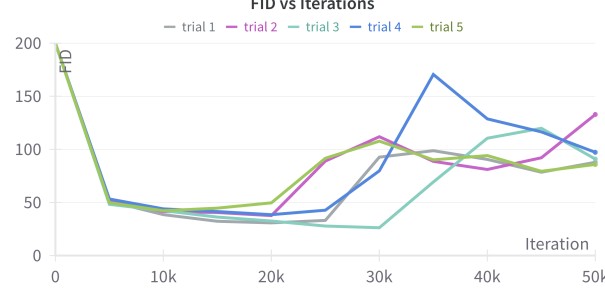

Figure 7: Convergence issues of `DDM-GAN`.

DDM-GAN in the sequel.

We observe such issue in the considered experiment (see Appendix B.5 for results). As we mentioned, such hardness in adversarial optimization is a reason motivating our DFM approach. Therefore, we present the best FID attained by DDM-GAN in the sequel.

**Result.** Fig. 8 shows the qualitative results of translation obtained by all methods. One can see that `CycleGAN`, `FM`, and `FM-OT`, which does not use the auxiliary information of gender, suffer from content misalignment issue. `DDM-GAN` does not show satisfiable content alignment either, probably due to the numerical instability as demonstrated in Fig. 7. `FM-cond`, the naive FM based implementation of DDM using linear interpolants in (9), show better content alignment than the other baselines (e.g., see first row). Nonetheless, `DFM` shows the best alignment, supporting the translation

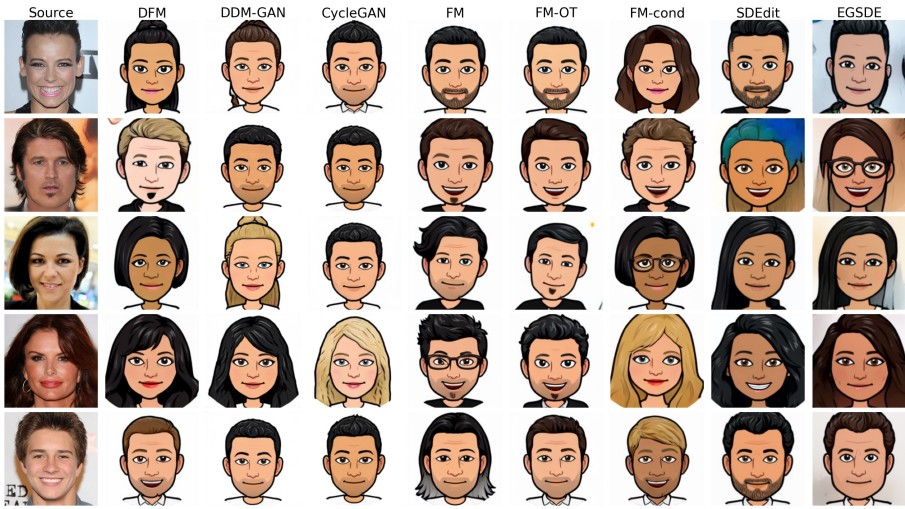

Figure 8: Human faces to Bitmoji translation by all methods.

Table 2: Quantitative results on the Face to Bitmoji task.

| Method | FID | DreamSim | Train Time (hrs) | Infer. Time (s) |
|---|---|---|---|---|
| *GAN-based* | | | | |
| DDM-GAN (Shrestha & Fu, 2024) | 26.20 | 0.58 (0.05) | 13.60 | **0.011** |
| CycleGAN (Zhu et al., 2017) | 31.14 | 0.63 (0.06) | 22.12 | **0.011** |
| *Diffusion-based* | | | | |
| SDEdit (k=3) | 63.37 | 0.62 (0.06) | 27.10 | 27.22 |
| EGSDE (k=3) | 69.93 | 0.56 (0.06) | 31.77 | 65.82 |
| *FM-based* | | | | |
| FM (Albergo et al., 2023) | 43.23 | 0.66 (0.06) | **3.08** | 5.28 |
| FM-cond (w/ Eq. 9) | 22.65 | 0.60 (0.06) | 3.15 | 5.30 |
| FM-OT (Tong et al., 2023) | 44.60 | 0.66 (0.06) | 12.42 | 5.21 |
| DFM (Ours) | **22.20** | **0.59** (0.05) | 6.17 | 5.27 |

identifiability claims of DDM.

Table 2 presents the quantitative results of all methods. FID assesses the distributional similarity between the translated and target images, while DreamSim evaluates content alignment between the source and translated images. The proposed method achieves the best balance between FID and DreamSim, indicating that it preserves high-quality domain translation without sacrificing content consistency.

### 5.3. Swarm Navigation

We also test our algorithm on an interesting robot swarm navigation problem (Liu et al., 2018; 2023a); please refer to Appendix C.1 for details.

## 6. Conclusion

In this work, we introduced DFM, a computational framework that integrates FM into the DDM criterion. DDM is a powerful criterion for UDT, which provably attains translation identifiability in UDT, solving content misalignment issues. DDM had only been realized by GANs, encountering numerical instability and losing transport trajectory

information. This motivated us to design a FM-based DDM approach. We first revealed the unique challenges of imposing DDM-induced constraints on flows. Then, we designed a bilevel formulation with private learnable nonlinear interpolants, provably recovering the DDM criterion using FM. We also provided an efficient two-stage implementation, avoiding computational barriers. Experiments demonstrated that DFM effectively computes the DDM criterion, serving as the first translation identifiability-guaranteed flow model.

**Limitations.** First, our UDT framework is restricted to one-to-one translations, whereas many applications, such as image translation, can benefit from one-to-many mappings. Extending FM-based methods to enable identifiable one-to-many translations is a promising yet challenging direction. Second, our efficient computing scheme relies on non-overlapping supports of the conditional distributions. For overlapped cases, how to efficiently realize the bilevel loss is worth considering in the future.

## Acknowledgment

This work was supported in part by the National Science Foundation CAREER Award ECCS-2144889.

## Impact Statement

This paper presents work whose goal is to advance the field of Machine Learning. There are many potential societal consequences of our work, none which we feel must be specifically highlighted here.

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

# Supplementary Material of "Diversified Flow Matching with Translation Identifiability"

## A. Notation

1. $\partial_t \boldsymbol{f}(t, \cdot)$ represents the partial derivative of $\boldsymbol{f}$ with respect to $t$.

2. $[\boldsymbol{f}]_{\#p}$ and $\boldsymbol{f}_{\#p}$ represents the push-forward of the density $p$ by the map $\boldsymbol{f} : \mathcal{X} \rightarrow \mathcal{Y}$, i.e., it satisfies $\boldsymbol{f}_{\#p}(\mathcal{A}) = p(\boldsymbol{f}^{-1}(\mathcal{A}))$ for a measruable set $\mathcal{A} \subseteq \mathcal{Y}$, where $\boldsymbol{f}^{-1}$ is the pre-image of $\boldsymbol{f}$ and $p(\mathcal{B})$ denotes the measure of set $\mathcal{B}$ under the distribution specified by density $p$.

3. $I_d \in \mathbb{R}^{d \times d}$ represents the identity matrix.

4. $\partial_t \boldsymbol{f}_t$ represents the partial derivative of $\boldsymbol{f}_t$ with respect to $t$.

5. $\nabla \cdot \boldsymbol{v}$ represents the divergence of vector field $\boldsymbol{v}$.

## B. Details on Proposed method and Challenges

### B.1. Robust Identifiability (Shrestha & Fu, 2024)

**Theorem B.1.** *(Shrestha & Fu, 2024) Let $\widehat{\boldsymbol{g}}$ be any optimal solution of the DDM criterion* (3). *Let* $\mathrm{dia}(\mathcal{A}) = \sup_{\boldsymbol{w}, \boldsymbol{z} \in \mathcal{A}} \|\boldsymbol{w} - \boldsymbol{z}\|_2$ *measure the size of a set. Let* $\mathcal{V} = \{(\mathcal{A}, \mathcal{B}) \mid SDC \text{ is violated on } (\mathcal{A}, \mathcal{B})\}$. *Assume that $\boldsymbol{g}^\star$ is an $L$-Lipchitz continuous function and that* $\max_{(\mathcal{A}, \mathcal{B}) \in \mathcal{V}} \max(\mathrm{dia}(\mathcal{A}), \mathrm{dia}(\mathcal{B})) \leq r$. *Then,*

$$\|\widehat{\boldsymbol{g}}(\boldsymbol{x}) - \boldsymbol{g}^\star(\boldsymbol{x})\|_2 \leq 2rL, \quad \forall \boldsymbol{x} \in \mathcal{X}.$$

That is, $\boldsymbol{g}^\star$ is identified to reasonable accuracy by the DDM criterion if the SDC holds approximately. Morevoer, the translation error increases only linearly with the size of sets in which the SDC condition is violated.

### B.2. Algorithm DFM

---
**Algorithm 1 DFM**

---
**Require:** $\rho, \sigma_1, \sigma_2$, initialized parameters $\phi, \boldsymbol{\theta}$
1: **while** Stopping criterion for $\boldsymbol{I}_{\boldsymbol{\theta}}$ is not met **do**
2:      Sample $(\boldsymbol{x}^{(q)}, \boldsymbol{y}^{(q)}) \sim \rho(\boldsymbol{x}, \boldsymbol{y}|u^{(q)})$ and $t_q \sim \mathcal{U}(0, 1), \forall q \in [Q]$
3:      $\boldsymbol{z}_{t_q, \boldsymbol{\theta}}^{(q)} \leftarrow (1 - t_q)x^{(q)} + t_q \boldsymbol{y}^{(q)} + t_q(1 - t_q)\boldsymbol{\gamma}_{\boldsymbol{\theta}}(\boldsymbol{x}^{(q)}, \boldsymbol{y}^{(q)}), \forall q \in [Q]$
4:      $\mathcal{L}_{\mathrm{interp}}(\boldsymbol{\theta}) \leftarrow \sum_{i=1}^{Q-1} \sum_{j=i+1}^{Q} \boldsymbol{\gamma}_{\sigma_2}(|t_i - t_j|)\boldsymbol{\gamma}_{\sigma_1}(\|\boldsymbol{z}_{t_i, \boldsymbol{\theta}}^{(i)} - \boldsymbol{z}_{t_j, \boldsymbol{\theta}}^{(j)}\|_2)$
5:      $\boldsymbol{\theta} \leftarrow \boldsymbol{\theta} - \alpha_1 \overline{\nabla}_{\boldsymbol{\theta}} \mathcal{L}_{\mathrm{interp}}(\boldsymbol{\theta})$
6: **end while**
7: **while** Stopping criterion for $v_t(\cdot; \phi)$ is not met **do**
8:      Sample $(\boldsymbol{x}, \boldsymbol{y}) \sim \rho(\boldsymbol{x}, \boldsymbol{y})$ and $t \sim \mathcal{U}(0, 1)$
9:      $\boldsymbol{z}_t \leftarrow (1 - t)\boldsymbol{x} + t\boldsymbol{y} + t(1 - t)\boldsymbol{\gamma}_{\boldsymbol{\theta}}(\boldsymbol{x}, \boldsymbol{y})$
10:      $\ell(\boldsymbol{\phi}) = \|\boldsymbol{v}_t(\boldsymbol{z}_t) - \partial_t \boldsymbol{z}_t\|_2^2$
11:      $\boldsymbol{\phi} \leftarrow \boldsymbol{\phi} - \alpha_2 \overline{\nabla}_{\boldsymbol{\phi}} \ell(\boldsymbol{\phi})$
12: **end while**
13: **return** $\boldsymbol{v}_t(\cdot; \boldsymbol{\phi})$

---

In Algorithm 1, $\overline{\nabla}$ represents gradient-based update direction. The specific optimizers used in the experiments are described in their corresponding experiment sections.

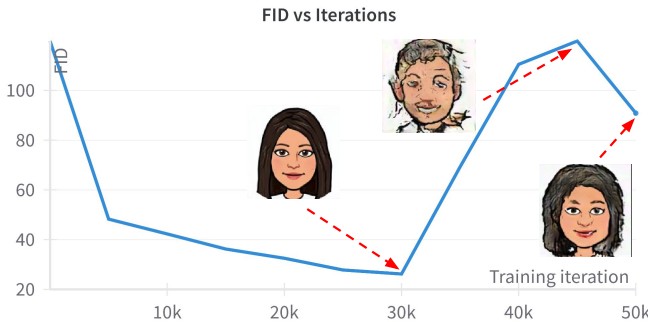

Figure 9: Image quality during the GAN training when convergence issues were encountered.

## B.3. Derivation of $\widehat{v}_{\frac{1}{2}}(\frac{1}{2}x + \frac{1}{2}y)$ in Fig. 3

If the vector field $v_t(z)$ is a solution to (8), then $v_t$ can be expressed as follows:

$$\widehat{v}_t(z) = \mathop{\mathbb{E}}_{\substack{u^{(q)} \sim p_u, x \sim p_{x|u^{(q)}} \\ y \sim p_{y|u^{(q)}}}} \left[ \partial_t I^{\text{linear}}(x, y, t) \mid I^{\text{linear}}(x, y, t) = z \right]$$

$$= \mathop{\mathbb{E}}_{\substack{u^{(q)} \sim p_u, x \sim p_{x|u^{(q)}} \\ y \sim p_{y|u^{(q)}}}} \left[ y - x \mid (1 - t)x + ty = z \right]$$

$$\widehat{v}_{\frac{1}{2}}(z) = \mathbb{E}_{u^{(q)} \sim p_u, x \sim p_{x|u}} [2z - 2x]$$

$$= \mathbb{E}_{x \sim p_x} [2z - 2x]$$

$$= 2(z - \mathbb{E}[x])$$

Hence $\widehat{v}_{\frac{1}{2}}$ points towards the mean of the two clusters, $p_{x|u^{(q)}}, q = 1, 2$, resulting in reflection.

### B.4. Pitfalls of Problem (23)

In order to evaluate how well Problem (23) works in practice, we run simulations on 2D and 3D synthetic data as presented in Fig. 4. We observed that directly using Problem (23) is challenging from optimization perspective, since the unified vector field need to minimize $\mathcal{L}_{\text{FM}}$ simultaneously with respect to all conditional distributions. Instead the following variable splitting based implementation that decouples $\mathcal{L}_{\text{FM}}$ loss minimization and vector field unification was found to be optimization friendly:

$$\underset{v_t, \{\widetilde{v}_t^{(q)}, \theta_q\}_{q=1}^Q}{\text{minimize}} \sum_{q=1}^{Q} \mathcal{L}_{\text{FM}} \left( \widetilde{v}_t^{(q)}, I_{\theta_q}^{(q)}, \rho(x, y | u^{(q)}) \right) + \lambda \mathbb{E}_{t, z_t \sim \rho_t^{(q)}} \left\| v_t(z_t) - \widetilde{v}^{(q)}(z_t) \right\|_2^2, \tag{18}$$

where $z_t \sim \rho_t^{(q)}$ implies $z_t = I_{\theta_q}^{(q)}(x, y, t), (x, y) \sim \rho(x, y | u^{(q)})$.

**Experiment Details.** The 2D Gaussian blobs for $p_{x|u_1}$ and $p_{x|u_2}$ for Fig. 4 are generated with unit variance at locations $(1, 1)$ and $(1, -1)$. $y = -x$ is used to generate $p_{y|u_1}$ and $p_{y|u_2}$. For 3D plot, the location of the blobs are $(1, 1, 0)$ and $(1, -1, 0)$.

### B.5. Convergence issue of GAN-based UDT

Fig. 9 shows an example training trail of `DDM-GAN` for the setting considered in Sec. 5.2 along with the sample translated images. It shows that once the convergence issues are encountered, the translation quality is unacceptable and can be considered as an optimization failure.

# C. Additional Experiments

## C.1. Swarm Navigation

Navigating a large swarm of robots, on land or in air, requires computing the control policies (i.e., the velocities) for individual robots that meet the specifications for their collective behavior (Liu et al., 2018). CNF-based methods can be used to recover individual robot control policy based on the robot's location and time such that the entire swarm navigates from source location to the destination (Liu et al., 2023a; Kapuśniak et al., 2024; Liu et al., 2018).

**Problem Description.** We consider swarm navigation on a complex land surface. The surface is specified by LiDAR measurements of Mt. Rainier (Legg & Anderson, 2013) containing 34,183 points. We consider a challenging scenario where there are two different swarms of robots with their own source and destination locations on the surface as shown in Fig. 10[Column 1]. We consider the source locations of the first and second swarm distributed as $p_{\boldsymbol{x}|u_1}$ and $p_{\boldsymbol{x}|u_2}$, and the destinations distributed as $p_{\boldsymbol{y}|u_1}$ and $p_{\boldsymbol{y}|u_2}$, respectively. Then, the goal is to transport samples from $p_{\boldsymbol{x}|u^{(q)}}$ to $p_{\boldsymbol{y}|u^{(q)}}, \forall q$, which aligns with the objective of DFM. This problem can be understood as a DDM problem. We want to control the clusters of robots so that they move from source to destination clusters, while avoiding collision among the clusters.

**Neural Networks and Hyperparameter settings.** We use a 3-layer MLP with 64 hidden units and SeLU activation to represent both $\boldsymbol{I_\theta}$ and $\boldsymbol{v}_t(\cdot; \boldsymbol{\phi})$. We use Adam optimizer for both interpolant and the vector field with an initial learning rate of $10^{-4}$ and $10^{-3}$ respectively. We use a weight decay of $10^{-5}$ for both networks. We use $\sigma_1 = 0.1, \sigma_2 = 1.5$.

**Regularization for surface adherence** In order to encourage the swarm to stay close to the land surface, we add a regularization from (Kapuśniak et al., 2024) in the first phase of our method, i.e., interpolant training in Algorithm 1. Since Problem (12) itself does not encourage trajectories to stay close to the land surface represented by LiDAR measurements, we use additional regularization from (Kapuśniak et al., 2024) that encourages trajectories to stay close to the surface. To explain, let $\mathcal{D} = \{m_i\}_{i=1}^N$ be the set of LiDAR measurements. Then, Line 4 and 5 in Algorithm 1 is modified as follows:

$$\ell(\boldsymbol{\theta}) = \lambda_1 \mathcal{L}_{\text{interp}}(\boldsymbol{\theta}) + \lambda_2 \mathcal{L}_{\text{mfm}}(\boldsymbol{\theta}),$$

where

$$\mathcal{L}_{\text{mfm}}(\boldsymbol{\theta}) = \left(\partial_t \boldsymbol{z}_{t_q,\theta}^{(q)}\right)^\top \boldsymbol{G}(\boldsymbol{z}_{t_q,\theta}^{(q)}, \mathcal{D}) \left(\partial_t \boldsymbol{z}_{t_q,\theta}^{(q)}\right),$$

where $\boldsymbol{G}(\boldsymbol{z}_{t_q,\theta}^{(q)}, \mathcal{D})$ is a data dependent metric introduced in (Kapuśniak et al., 2024) to "pull" the interpolant paths closer to the manifold represented by $\mathcal{D}$.

For our experiments Sec. 5.3, we use "LAND" originally introduced in Arvanitidis et al. (2016) (see details in (Kapuśniak et al., 2024)), with hyperparameter $\sigma = 0.125$. We set $\lambda_1 = 5000$ and $\lambda_2 = 1$.

**Dataset.** In the experiment in Fig. 10, we use Gaussians to represent $p_{\boldsymbol{x}|u^{(q)}}$ and $p_{\boldsymbol{y}|u^{(q)}}$. We use a variance of 0.02 for $p_{\boldsymbol{x}|u^{(q)}}$ and that of 0.03 for $p_{\boldsymbol{y}|u^{(q)}}, \forall q \in \{1, 2\}$. We use $K = 4000$ samples for each of the conditional distributions.

**Metric.** We use surface adherence (SA) metric to measure how close the trajectory is on average to the surface specified by LiDAR measurements. `SA` is defined as follows:

$$\text{SA} = \frac{1}{N} \sum_{n=1}^N \sum_{\tau=1}^T |[\boldsymbol{x}_\tau]_3 - [\text{NN}([\boldsymbol{x}_\tau]_{1:2}; \mathcal{D}_{1:2})]_3|, \tag{19}$$

where $\text{NN}([\boldsymbol{x}_\tau]_{1:2}; \mathcal{D}_{1:2})$ represents the nearest neighbor of $\boldsymbol{x}_\tau$ in the set $\mathcal{D}$ while only considering the first and the second coordinates ,i.e., $x, y$ location without the height.

**Result.** Fig. 10 shows the trajectories obtained by the proposed method and the baselines. One can see that the trajectories returned by the baselines almost overlap for different swarms, and poorly adhere to the land surface (e.g., crossing underneath the mountain) However, the proposed `DFM` method returns different trajectories for different swarms while staying closely to the surface.

Table 3 shows the SA obtained by all methods averaged over 5 trials. Combined observation from Table 3 and Fig. 10 shows that the proposed method shows better swarm navigation, in terms of transport and surface adherence, compared to the baselines. Note that the color in Fig. 10 indicates the time $t$. This experiment also shows that `DFM` is useful in tasks requiring simultaneous trajectory estimation between multiple pairs of distributions.

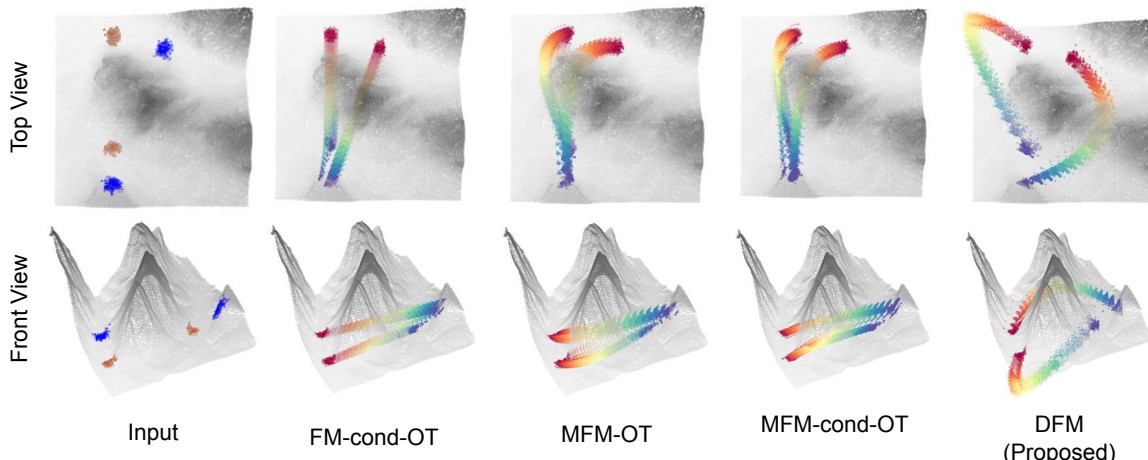

Figure 10: Visualization of the trajectory obtained by different methods.[Column 1 Top] source locations of two swarms (red and blue) on the top and corresponding destinations on the bottom of the plot. [Column 2 - 5] swarm trajectories obtained by different methods, color specify the time. From red to blue, $t = 0$ to $1$.

Table 3: Average SA with standard deviation for different methods.

| Method | SA |
|---|---|
| FM-OT (Tong et al., 2023) | $1.09 \pm 0.027$ |
| FM-cond-OT (Tong et al., 2023) w/ (9) | $1.08 \pm 0.019$ |
| MFM-OT (Kapuśniak et al., 2024) | $0.47 \pm 0.015$ |
| MFM-cond-OT (Kapuśniak et al., 2024) w/ (9) | $0.43 \pm 0.055$ |
| DFM (Proposed) | $\mathbf{0.32 \pm 0.080}$ |

## D. Experiment Details

### D.1. Unpaired Image to Image Translation

**Neural Networks and Hyperparameter settings.** We use the UNet architecture (Ronneberger et al., 2015) to represent both neural networks. We adopt a similar hyperparameter configuration based on the UNet architecture (Dhariwal & Nichol, 2021). For the vector field, we use the AdamW optimizer (Loshchilov, 2017) with an initial learning rate of $10^{-4}$ and parameters $\beta_1 = 0.9$, $\beta_2 = 0.999$, $\epsilon = 1e - 8$, and no weight decay. We use a batch size of 64, and dropout of 0.1. We take the exponential moving average (EMA) (Tarvainen & Valpola, 2017) of the weights with decay parameter 0.9999. We use the same hyperparameter settings for the interpolant, except that the learning rate is set to $10^{-8}$, head channels is 32 and attention resolution is 8. We use $\sigma_1 = 0.1$ and $\sigma_2 = 10$. We train the models for 100k iterations. All baselines are also trained for the same number of iterations.

For the image translation experiment, we observed improved training stability and efficiency when modifying Algorithm 1 so that $\boldsymbol{\theta}$-updates and $\boldsymbol{\phi}$-updates are interleaved, resulting in Algorithm 2. The primary reason is that learning the vector field in the second phase of Algorithm 1 with respect to a complex nonlinear interpolant appears challenging, as evidenced by oscillations in the objective function. However, training the vector field and interpolant in tandem alleviates this issue, as the randomly initialized $\boldsymbol{I_\theta}$ at the beginning of training is close to a linear interpolant. This results in a more gradual training of vector field from almost linear to increasingly nonlinear paths.

## E. Proofs

### E.1. Proposition 3.4

To prove Lemma 3.4, first consider the following definition:

Table 4: Hyperparameters of the UNets for unpaired image translation task.

| Hyperparameter | Value |
|---|---|
| Attention resolution | 16 |
| Heads channels | 64 |
| Heads | 1 |
| Channels multiple | 2, 2, 2 |
| ResNet blocks | 4 |
| Channels | 128 |

---

**Algorithm 2 DFM with Interleaved Training**

---

**Require:** $\rho, \sigma_1, \sigma_2$, initialized parameters $\phi, \boldsymbol{\theta}$

1: **while** Stopping criterion is not met **do**
2:    Sample $(\boldsymbol{x}^{(q)}, \boldsymbol{y}^{(q)}) \sim \rho(\boldsymbol{x}, \boldsymbol{y}|u^{(q)})$ and $t_q \sim \mathcal{U}(0, 1), \forall q \in [Q]$
3:    $\boldsymbol{z}_{t_q,\boldsymbol{\theta}}^{(q)} \leftarrow (1 - t_q)x^{(q)} + t_q \boldsymbol{y}^{(q)} + t_q(1 - t_q)\boldsymbol{\gamma}_{\boldsymbol{\theta}}(\boldsymbol{x}^{(q)}, \boldsymbol{y}^{(q)}), \forall q \in [Q]$
4:    $\mathcal{L}_{\mathrm{interp}}(\boldsymbol{\theta}) \leftarrow \sum_{i=1}^{Q-1} \sum_{j=i+1}^{Q} \boldsymbol{\gamma}_{\sigma_2}(|t_i - t_j|)\boldsymbol{\gamma}_{\sigma_1}(\|\boldsymbol{z}_{t_i,\boldsymbol{\theta}}^{(i)} - \boldsymbol{z}_{t_j,\boldsymbol{\theta}}^{(j)}\|_2)$
5:    $\boldsymbol{\theta} \leftarrow \boldsymbol{\theta} - \overline{\nabla}_{\boldsymbol{\theta}}\mathcal{L}_{\mathrm{interp}}(\boldsymbol{\theta})$
6:    $\mathcal{L}_{\mathrm{FM}}(\boldsymbol{\phi}) \leftarrow \frac{1}{Q} \sum_{q=1}^{Q} \|\boldsymbol{v}_t(\boldsymbol{z}_{t_q,\boldsymbol{\theta}}^{(q)}) - \partial_t \boldsymbol{z}_{t_q,\boldsymbol{\theta}}^{(q)}\|_2^2$
7:    $\boldsymbol{\phi} \leftarrow \boldsymbol{\phi} - \overline{\nabla}_{\boldsymbol{\phi}}\mathcal{L}_{\mathrm{FM}}(\boldsymbol{\phi})$
8: **end while**
9: **return** $\boldsymbol{v}_t(\cdot; \boldsymbol{\phi})$

---

**Definition E.1** (Interpolable density, Albergo et al. (2023) Def. D.1). A probability path $p_t$ with $t \in [0, 1]$ is interpolable if there exists a time dependent invertible map $\boldsymbol{\psi}_t : \mathbb{R}^d \to \mathbb{R}^d$ with $t \in [0, 1]$, continuously-differentiable in time and space, such that $p_t$ is the pushforward of $\boldsymbol{\psi}_t$ of the standard normal density, i.e., $[\boldsymbol{\psi}_t]_{\#\mathcal{N}(0,I)} = p_t$.

This definition will be helpful in proving Proposition 3.4

> **Proposition 3.4.** Suppose that there exists a flow $\boldsymbol{f}_t^\star : \mathbb{R}^d \to \mathbb{R}^d$, differentiable in time and space, from $p_{\boldsymbol{x}}$ to $p_{\boldsymbol{y}}$ such that $\boldsymbol{f}_1^\star = \boldsymbol{g}^\star$. Suppose there exist a diffeomorphism from a $\mathcal{N}(0, I_d)$ to $p_{\boldsymbol{x}|u^{(q)}}, \forall q$. Let $\widehat{\boldsymbol{v}}_t$ denote a solution of Problem (12) and denote $\boldsymbol{g}_{\widehat{\boldsymbol{v}}}(\boldsymbol{x}) = \boldsymbol{x} + \int_{t=0}^{t=1} \widehat{\boldsymbol{v}}_t(\boldsymbol{z}_t)dt$. Then, under the same model in (1), when $\{p_{\boldsymbol{x}|u^{(q)}}\}_{q=1}^{Q}$ satisfies the SDC, we have $\boldsymbol{g}_{\widehat{\boldsymbol{v}}} = \boldsymbol{g}^\star$, holds a.e.

*Proof.* Let $\psi_0^{(q)}$ denote a deffeomorphism from $\mathcal{N}(0, I_d)$ to $p_{\boldsymbol{x}|u^{(q)}}$. Further, let $\psi_t^{(q)} = \boldsymbol{f}_t^\star \circ \psi_0^{(q)}$. This implies that $[\psi_0^{(q)}]_{\#\mathcal{N}(0,I)} = p_{\boldsymbol{x}|u^{(q)}}$ and $[\psi_1^{(q)}]_{\#\mathcal{N}(0,I)} = p_{\boldsymbol{y}|u^{(q)}}$. Hence, $\psi_t^{(q)}$ is a time and space continuously-differntiable invertible map from $p_{\boldsymbol{x}|u^{(q)}}$ to $p_{\boldsymbol{y}|u^{(q)}}$.

Moreover, let $p_t^{(q)} = [\psi_t^{(q)}]_{\#p_{\boldsymbol{x}|u^{(q)}}}$. Then $p_t^{(q)}, \forall q$ are interpolable densities in the sense of Definition E.1.

Now, consider the following result from (Albergo et al., 2023)

**Proposition E.2** ((Albergo et al., 2023) Proposition D.1). *Let $\rho_t$ be an interpolable density in the sense of Definition E.1 with corresponding map $\boldsymbol{\psi}_t$, i.e., $[\boldsymbol{\psi}_t]_{\#\mathcal{N}(0,I)} = \rho_t$. Then*

$$\boldsymbol{I}(\boldsymbol{x}, \boldsymbol{y}, t) = \boldsymbol{\psi}_t\Big(\boldsymbol{\psi}_0^{-1}(\boldsymbol{x}) \cos\big(\tfrac{\pi}{2}t\big) + \boldsymbol{\psi}_1^{-1}(\boldsymbol{y}) \sin\big(\tfrac{\pi}{2}t\big)\Big) \tag{20}$$

*is such that $\boldsymbol{I} \in \mathcal{I}$ for independent coupling, i.e., $\boldsymbol{x} \sim \rho_0$ and $\boldsymbol{y} \sim \rho_1$.*

Invoking the above proposition, we have that for each $q \in [Q]$, there exists an interpolant function that can follow the probability path $p_t^{(q)}$.

We use a similar reformulation technique as in Albergo et al. (2023)(Appendix D) to re-express our constraints as follows:

$$\min_{\boldsymbol{v}_t, \rho_t^{(q)}, \boldsymbol{v}_t^{(q)}} \sum_{q=1}^{Q} \|\boldsymbol{v}_t^{(q)} - \boldsymbol{v}_t\|^2 \tag{21a}$$

$$\text{subject to}: \ \partial_t \rho_t^{(q)} + \nabla \cdot (\boldsymbol{v}_t^{(q)} \rho_t^{(q)}) = 0, \tag{21b}$$

$$\rho_0^{(q)} = p_{\boldsymbol{x}|u^{(q)}}, \rho_1^{(q)} = p_{\boldsymbol{y}|u^{(q)}} \tag{21c}$$

$$\rho_t^{(q)} \text{ interpolable,} \tag{21d}$$

where $\nabla \cdot \boldsymbol{u}$ represents the divergence of vector field $\boldsymbol{u}$, i.e., $\nabla \cdot \boldsymbol{u}(\boldsymbol{x}) = \sum_{i=1}^{n} \frac{\partial u_i}{\partial x_i}$. Here, the optimization is over the probability path $\rho_t^{(q)}$ instead of the interpolants. Note that the constraints (21c) (21d) restricts the search over probability paths that can be represented by interpolants in $\mathcal{I}$. In other words, $\rho_t^{(q)}$ in Problem (21) is a reparametrization of $\boldsymbol{I}^{(q)}$ in Problem (12). Finally, the constraint (21b) is the continuity equation that ensures that the private vector fields $\boldsymbol{v}_t^{(q)}$ follow the probability path $\rho_t^{(q)}$ (Villani et al., 2009; Lipman et al., 2023; Albergo et al., 2023). Therefore, (21b) is equivalent to (12b).

Let $\boldsymbol{v}_t^\star(\boldsymbol{f}^\star(\boldsymbol{x})) = \partial_t \boldsymbol{f}_t^\star(\boldsymbol{x})$. Then, $\boldsymbol{v}^\star, p_t^{(q)}, \forall q$ constitutes a feasible solution to Problem (21) that can attain zero objective. This implies that Problem (12) can also attain an objective of zero at the minimum.

Let $\widehat{\boldsymbol{v}}^{(q)}$ and $\widehat{\boldsymbol{I}}^{(q)} \in \mathcal{I}$ denote the private vector fields and interpolants returned by solving Problem (12). Note that when the constraint (12b) is satisfied, $\widehat{\boldsymbol{v}}^{(q)}$ transports $p_{\boldsymbol{x}|u^{(q)}}$ to $p_{\boldsymbol{y}|u^{(q)}}$ (Albergo et al., 2023).

Since there exists at least one solution to Problem (12) which attains zero objective value, $\widehat{\boldsymbol{v}}_t$ must also satisfy

$$\sum_{q=1}^{Q} \|\widehat{\boldsymbol{v}}_t - \widehat{\boldsymbol{v}}_t^{(q)}\|_2 = 0$$
$$\implies \widehat{\boldsymbol{v}}_t = \widehat{\boldsymbol{v}}_t^{(q)}, \forall q \in [Q]$$

Hence $\widehat{\boldsymbol{v}}_t$ is a unified vector field that transports $p_{\boldsymbol{x}|u^{(q)}}$ to $p_{\boldsymbol{y}|u^{(q)}}, \forall q \in [Q]$.

Therefore, $\boldsymbol{g}_{\widehat{\boldsymbol{v}}}$ satisfies

$$[\boldsymbol{g}_{\widehat{\boldsymbol{v}}}]_{\#p_{\boldsymbol{x}|u}} = p_{\boldsymbol{y}|u}$$

Hence $\boldsymbol{g}_{\widehat{\boldsymbol{v}}}$ is a solution to Problem (4). Therefore, invoking Theorem 2.2, $\boldsymbol{g}_{\widehat{\boldsymbol{v}}} = \boldsymbol{g}^\star$, a.e. $\qquad \square$

### E.2. Proof of Fact 3.3

Let us define the solution sets $\mathcal{V}^{(q)}$ as follows:

$$\mathcal{V}^{(q)} = \{\boldsymbol{v}_t \mid \boldsymbol{v}_t \text{ transports } p_{\boldsymbol{x}|u^{(q)}} \text{ to } p_{\boldsymbol{y}|u^{(q)}}\}$$
$$= \left\{\boldsymbol{v}_t \mid \boldsymbol{v}_t = \arg\min_{\boldsymbol{w}_t} \mathcal{L}_q(\boldsymbol{w}_t, \boldsymbol{I}), \forall \boldsymbol{I} \in \mathcal{I}\right\},$$

where $\mathcal{L}_q = \mathcal{L}_{\text{FM}}(\cdot, \cdot, \rho(\boldsymbol{x}, \boldsymbol{y}|u^{(q)}))$ for brevity. Note that $\mathcal{V}^{(q)}$ is the set of all vector fields that can transport $p_{\boldsymbol{x}|u^{(q)}}$ to $p_{\boldsymbol{y}|u^{(q)}}$. It is clear that $\mathcal{V}^{(i)} \neq \mathcal{V}^{(j)}$ for $i \neq j$ in general. Our goal is to find a unified vector field that can simultaneously transport $p_{\boldsymbol{x}|u^{(q)}}$ to $p_{\boldsymbol{y}|u^{(q)}}, \forall q$. To that end, let

$$\mathcal{V} = \{\boldsymbol{v}_t \mid \boldsymbol{v}_t \text{ transports } p_{\boldsymbol{x}|u^{(q)}} \text{ to } p_{\boldsymbol{y}|u^{(q)}}, \forall q\}$$
$$= \{\boldsymbol{v}_t \mid \boldsymbol{v}_t \in \mathcal{V}^{(1)} \ \& \ \dots \ \& \ \boldsymbol{v}_t \in \mathcal{V}^{(Q)}\}$$
$$= \cap_{q=1}^{Q} \mathcal{V}^{(q)}. \tag{22}$$

For the ease of exposition, let $Q = 2$. Our argument will generalize to arbitrary $Q$. Let

$$\overline{\boldsymbol{v}}_t^{(1)}, \overline{\boldsymbol{I}}^{(1)} = \arg \min_{\boldsymbol{v}_t, \boldsymbol{I} \in \mathcal{I}} \mathcal{L}_q(\boldsymbol{v}_t, \boldsymbol{I}) \quad \text{and}$$

$$\widehat{v}_t, \widehat{\boldsymbol{I}}^{(1)}, \widehat{\boldsymbol{I}}^{(2)} = \arg \min_{\substack{\boldsymbol{v}_t \in \mathcal{V}, \\ \boldsymbol{I}^{(1)}, \boldsymbol{I}^{(2)} \in \mathcal{I}}} \sum_{q=1}^{2} \mathcal{L}_q(\boldsymbol{v}_t, \boldsymbol{I}^{(q)}), \tag{23}$$

Note that in Problem (23) $\boldsymbol{v}_t \in \mathcal{V}$ is explicitly enforced. Whereas in Problem (10), $\boldsymbol{v}_t \in \mathcal{V}$ is not enforced but is what we hope for.

Nonetheless the definition of $\overline{\boldsymbol{v}}_t^{(1)}, \overline{\boldsymbol{I}}^{(1)}$ implies that

$$\mathcal{L}_1(\overline{\boldsymbol{v}}_t^{(1)}, \overline{\boldsymbol{I}}^{(1)}) \le \mathcal{L}_1(\widehat{\boldsymbol{v}}_t^{(1)}, \widehat{\boldsymbol{I}}^{(1)}).$$

However, if $\overline{\boldsymbol{v}}_t^{(1)} \notin \mathcal{V}$ and $\mathcal{L}_1(\overline{\boldsymbol{v}}_t^{(1)}, \overline{\boldsymbol{I}}^{(1)}) \ll \mathcal{L}_1(\widehat{\boldsymbol{v}}_t^{(1)}, \widehat{\boldsymbol{I}}^{(1)})$, then there may exist $\widetilde{\boldsymbol{I}}^{(2)} \in \mathcal{I}$ such that

$$\mathcal{L}_1(\widehat{\boldsymbol{v}}_t, \widehat{\boldsymbol{I}}^{(1)}) - \mathcal{L}_1(\overline{\boldsymbol{v}}_t^{(1)}, \overline{\boldsymbol{I}}^{(1)}) > \mathcal{L}_2(\overline{\boldsymbol{v}}_t^{(1)}, \widetilde{\boldsymbol{I}}^{(2)}) - \mathcal{L}_2(\widehat{\boldsymbol{v}}_t, \widehat{\boldsymbol{I}}^{(2)})$$

$$\implies \mathcal{L}_1(\overline{\boldsymbol{v}}_t^{(1)}, \overline{\boldsymbol{I}}^{(1)}) + \mathcal{L}_2(\overline{\boldsymbol{v}}_t^{(1)}, \widetilde{\boldsymbol{I}}^{(2)}) < \mathcal{L}_1(\widehat{\boldsymbol{v}}_t, \widehat{\boldsymbol{I}}^{(1)}) + \mathcal{L}_2(\widehat{\boldsymbol{v}}_t, \widehat{\boldsymbol{I}}^{(2)}).$$

Since $(\overline{\boldsymbol{v}}_t^{(1)}, \overline{\boldsymbol{I}}^{(1)}, \widetilde{\boldsymbol{I}}^{(2)})$ is also a feasible solution to Problem (10), we can conclude that Problem (10) is not equivalent to Problem (23). The vector field returned by Problem (10) does not guarantee DDM satisfaction, since its solution may not be from $\mathcal{V}$.

