# OpenReview forum: "Diversified Flow Matching with Translation Identifiability"
_ICML.cc/2025/Conference — ICML 2025 poster_

### Official Review · Reviewer_dKp3 · 2025-03-08

**Overall Recommendation:** 3

**Summary:**

This paper proposed a flow matching model for diversified distribution matching, called DFM. The proposed method formulates a bilevel optimization problem to learn an interpolant and train a flow model on this interpolant. This paper demonstrates that the standard flow matching models fail in the DDM task. To address this, DFM proposes a method for learning a non-intersecting interpolant (Eq. 14) by leveraging the class label information. Intuitively, the objective encourages the trajectories of different classes to be well-separated. DFM model is evaluated on synthetic data, image-to-image translation, and swarm navigation (appendix).

**Claims And Evidence:**

- DFM model is supported by the experimental results.
- However, there is concern regarding the novelty of Prop 3.4, as it is mainly a restatement of the assumption that $f_{1}^{\star}=g^{\star}$ and Thm 2.2.

**Essential References Not Discussed:**

No

**Experimental Designs Or Analyses:**

- DFM model is evaluated on synthetic data, image-to-image translation, and swarm navigation (appendix).

- The image translation experiment is conducted on a relatively non-standard benchmark of CelebAHQ-to-Bitmoji. For reference, see [1, 2].

- There are concerns regarding the presentation. Most of the experimental results comparing DFM with baseline methods are placed in the appendix, including Fig 9, 10, 11, and Tab2. Additionally, all swarm navigation results are placed in the appendix.

- While the main target task is diversified distribution matching, the image translation experiments only evaluate marginal distributions, not conditional distribution matching. For reference, DDM-GAN measured the LPIPS score on a similar experiment [3].


[1] De Bortoli, Valentin, et al. "Schrodinger Bridge Flow for Unpaired Data Translation." NeurIPS 2024.
[2] Liu, Guan-Horng, et al. "I $^ 2$ SB: Image-to-Image Schr\" odinger Bridge." ICML 2023.
[3] Shrestha, Sagar, and Xiao Fu. "Towards identifiable unsupervised domain translation: A diversified distribution matching approach." ICLR 2024.

**Methods And Evaluation Criteria:**

- DFM aims to address the unsupervised domain traslation (UDT) problem. However, this model requires class labels for input-output data, raising concerns about whether this setting is also considered as UDT.

**Other Comments Or Suggestions:**

No

**Other Strengths And Weaknesses:**

**Strengths**
- The paper is well-written and easy to follow.
- The motivation is well-supported by the experiments in Section 3.2.

**Weakness**
- There are concerns regarding the necessity of Thm 2.3.
- The motivation section presents results that are somewhat expected, as observed in [1].
- Additional concerns are included in other sections.


[1] Liu, Xingchao, Chengyue Gong, and Qiang Liu. "Flow straight and fast: Learning to generate and transfer data with rectified flow."

**Questions For Authors:**

No

**Relation To Broader Scientific Literature:**

This paper proposed a flow matching model for the diversified distribution matching (DDM) task.

**Theoretical Claims:**

- There is concern regarding the novelty of Prop 3.4 (See **Claims And Evidence**).
- "+ $1_{\mathcal{I}}$" in Equation 12 may need to be replaced with "* $1_{\mathcal{I}}$".

---

> ### Author Rebuttal · Authors · 2025-04-01
>
> **[Novelty of Proposition 3.4]**
>
> The novelty lies in how to use FM-based losses to attain the same conclusion of Thm 2.2. Note that Thm 2.2 assumes that distribution matching is already attained. But Prop 3.4 specifically needs the distribution matching part to be realized by FM. How to achieve this while not violating the constraints in Thm 2.2 was very unclear before we came to the loss design of the bilevel objective. Coming to our design was entirely nontrivial and this constitutes the major novelty. We will articulate this point by adding a remark after Prop 3.4.
>
> &nbsp;
>
> **[“$ \times 1_{\mathcal{I}}$”  instead of “$+1\_{\mathcal{I}}$” ]**
>
> Thank you for your careful reading. We seemed to have missed the definition of the identity function $1_{\mathcal{I}}$ leading to confusion. The operator should be $+$ when
> $1_{\mathcal{I}}(x) = 0$ for $x \in \mathcal{I} $ and infinity otherwise. Whereas the operator should be $*$ when $1_{\mathcal{I}}(x) = 1$ for $x \in \mathcal{I} $ and infinity otherwise.
> We will include the definition to avoid this confusion.
>
> &nbsp;
>
> **[Presentation ]**
>
> Some experiments were moved to the appendix due to space constraints. To alleviate the reviewer’s concerns, we will restructure the paper to include partial results from all experiments along with baseline comparisons in the main paper and move the rest to the appendix.
>
> &nbsp;
>
> **[ LPIPS score ]**
>
> Image similarity metrics were not presented because we observed that metrics such as LPIPS do not make sense when the images have large domain gaps, which is the case in the considered experiment. For example, DFM and CycleGAN achieved the same LPIPS score, despite DFM being clearly better than CycleGAN in terms of alignment. Also note that LPIPS score was still not presented for the CelebAHQ to Bitmoji translation task in DDM-GAN .
>
> Nonetheless, to alleviate the reviewer’s concern, we evaluate the DreamSim [R1] score in the table below, which was shown to better align with human judgement than LPIPS score. Although the metric can provide some rough idea, it still cannot capture some obvious issues (e.g., DDM-GAN has diversity issues as seen in Fig. 7, last two rows, in the manuscript).
>
> *[R1] Fu, Stephanie, et al. "Dreamsim: Learning new dimensions of human visual similarity using synthetic data."*
>
>
> ------------------------------
>
> Method    $\quad\quad$      Mean (Std)
>
> ------------------------------
>
> DFM       $\quad\quad\quad~~$        0.59 (0.05)
>
> DDM-GAN   $\quad$   0.58 (0.05)
>
> CycleGAN  $\quad~$    0.63 (0.06)
>
> FM   $\quad\quad\quad~~~~$  0.66 (0.06)
>
> FM-OT  $\quad\quad~~$       0.66 (0.06)
>
> FM-cond    $\quad~~$    0.60 (0.06)
>
> SDEdit	$\quad\quad~~$       0.62 (0.06)
>
> ------------------------------
>
> We will add the above result in the revised version.
>
> &nbsp;
>
> **[Standard Benchmark Dataset]**
>
> Note that standard benchmark datasets (CelebAHQ Male to Female and AFHQ) can actually be considered as easy cases because the domain gap is relatively small. Hence, the issue of content misalignment was not observed by the existing methods [1,2]. For this reason, we intentionally did not include them, as they did not help make our point.
>
> However, in the case of CelebAHQ-to-Bitmoji, the content misalignment issue is clear, and therefore helps to prove our point.
>
> &nbsp;
>
> **[Necessity of Thm 2.3]**
>
> Indeed, we presented Thm 2.3 for showing the robustness of Thm 2.2. As Theorem 2.3 is not the contribution of this work, we only presented the theorem to provide sufficient context. However, we understand the reviewer’s concerns that Theorem 2.3 was not used explicitly in any of our own theoretical analysis, so might not be necessary to present it.
>
> We will move the theorem to the appendix to save space for other important revisions to the paper.
>
> &nbsp;
>
>
> [**Comparison with (Liu et al., 2023)**]
>
> Note that the example in the motivation is not claimed to be the contribution of our work. It is merely to illustrate the issue with using linear interpolants to the readers. Hence, we believe that a similar example showing reflecting trajectories does not undermine our work. Nonetheless, we understand that the example might be obvious for some readers familiar with the work. Hence we will cite [1] before elaborating on the example.
>
> &nbsp;
>
> **[ UDT terminology ]**
>
> The work can be considered unsupervised in that no paired data is required at all. However, we understand the reviewer’s concern and will consistently use “unpaired domain translation” for UDT instead of “unsupervised domain translation” in the revised version.

---

> > ### Comment · Reviewer_dKp3 · 2025-04-05
> >
> > Initially, I submitted an official comment that was not visible to the authors, so I am reposting it here:
> >
> > I appreciate the authors for the response and the additional experiments. Those have been helpful in addressing my concerns. Therefore, I will raise my rating to 3.

---

> > > ### Author Response · Authors · 2025-04-05
> > >
> > > Thank you for your response and valuable feedback to further refine our work.

---

### Official Review · Reviewer_Y3HP · 2025-03-12

**Overall Recommendation:** 3

**Summary:**

The paper introduces Diversified Flow Matching (DFM), a novel unsupervised domain translation (UDT) framework that extends ODE-based Flow Matching (FM) from linear to nonlinear interpolants, addressing the critical limitation of translation identifiability.

Key Contributions:

Overcoming Linear Flow Matching Limitations
- Prior FM approaches fail when distribution modes overlap, leading to incorrect mappings due to linear interpolants.
- The authors explicitly demonstrate this failure case with synthetic Gaussian mixtures where mode overlaps disrupt proper translation trajectories.

Advancing Nonlinear Flow Matching for Identifiability
- Builds on prior learnable flow models that incorporate nonlinear interpolants to improve flow-based translation.
- While previous works in metric-based flow modeling and Schrödinger bridge methods introduced learnable transport functions, they did not explicitly guarantee translation identifiability.
- The authors extend this concept by introducing higher-order non-intersecting nonlinear interpolants to explicitly enforce identifiability in domain translation.

Bilevel Optimization Loss for Enforcing Identifiability
- A novel bilevel learning loss is proposed, ensuring a unified, non-intersecting translation trajectory across different domain conditions.
- This approach is a unique theoretical and practical advancement, correcting mode overlap issues seen in prior FM methods.

**Claims And Evidence:**

(1) Claim 1: Linear Flow Matching Fails for Translation Identifiability

Evidence:

- Synthetic Gaussian mixture experiments clearly demonstrate incorrect trajectory mappings when using linear interpolants.
- Prior FM methods fail in mode-overlapping settings, leading to poor translation accuracy.

Limitation:

- Needs real-world validation in high-dimensional datasets.



(2) Claim 2: Bilevel Learning Loss Ensures Identifiability

Evidence:

- Proposition 3.4: Provides formal proof that the bilevel loss enforces translation identifiability by learning non-intersecting interpolants.

- Empirical results:

(a) Synthetic data: DFM avoids trajectory conflicts where FM-cond fails.
(b) Image translation: FID scores demonstrate improvements, but more qualitative analysis is needed to attribute gains specifically to translation identifiability.
(c) Swarm navigation: DFM preserves transport trajectories where standard FM methods fail.

Limitation:

- Table 1 lacks clarity → Needs explicit quantitative and qualitative validation that the observed FID improvements are due to translation identifiability rather than other architectural enhancements.

- Real-world experimental results lack qualitative and quantitative ablation studies on the effect of the bilevel optimization loss.
(a) No direct comparison is made between DFM with and without the bilevel loss in a real-world scenario.
(b) Translation identifiability should be evaluated explicitly in real-world tasks, particularly in structured medical or multimodal datasets where domain shifts are crucial.



(3) Claim 3: DFM Achieves Superior Content Alignment Over FM-cond and GAN-based DDM

Evidence:
- Face-to-Bitmoji translation experiments show better content alignment than FM-cond and GAN-based DDM.
- Table 2 (synthetic results) shows clear performance improvements, suggesting the effectiveness of DFM’s nonlinear flow matching.

Limitation:
- Table 1’s performance improvements require stronger justification.
- Need further ablation studies → Testing whether DFM’s gains persist without nonlinear interpolants.
- More real-world experiments needed to confirm generalization beyond synthetic tasks.

**Essential References Not Discussed:**

I have no clear reference to mention, but please refer to my comment on Relation to Broader Scientific Literature.

**Experimental Designs Or Analyses:**

I recommend that the author review and supplement the following parts:

(1) Ablation Study: Effect of the Bilevel Optimization Loss on Translation Identifiability
- Ablation experiments should test DFM with and without the bilevel loss to show how much it contributes to translation identifiability.
- Qualitative trajectory comparisons should illustrate the effect of the bilevel loss on mode separation and content preservation.
- Quantitative experiments should assess domain translation accuracy and alignment differences with and without the bilevel loss.

(2) Real-World Verification of Bilevel Learning Loss Effectiveness
- Demonstrate a real-world use case where translation identifiability affects model performance.
- Quantify performance improvement using classification accuracy, error reduction in domain adaptation tasks.

(3) Addressing Ambiguity Between Table 1 and Table 2 Results
- Table 2 (synthetic experiments) shows clearer performance improvements than Table 1, strongly supporting DFM’s effectiveness.
- Table 1’s real-world results, however, remain ambiguous, particularly regarding:
(a) Whether the observed performance difference between DFM and FM-cond is due to translation identifiability or other confounding factors. (b) Lack of qualitative trajectory visualizations and structured ablation to confirm that the bilevel optimization loss directly improves translation identifiability.

(4) Other suggestions
- Additional experiments aligning real-world results (Table 1) with synthetic evaluations (Table 2).
- Qualitative trajectory comparisons in real-world settings to explicitly visualize translation improvements.
- Ablation study breaking down the exact contributions of translation identifiability in Table 1 results.
- If real-world dataset expansion is infeasible, at least include additional qualitative and quantitative results in the appendix.

**Methods And Evaluation Criteria:**

Technical Approach's Contribution :

- Introduces a custom bilevel optimization loss with nonlinear, non-intersecting interpolants to explicitly enforce translation identifiability.
- Unlike prior FM-based methods, the bilevel optimization ensures that each interpolant follows a unified translation trajectory, avoiding trajectory conflicts.
- Simplifies the bilevel optimization problem by leveraging non-overlapping conditional distributions, making it computationally efficient.

Evaluation Metrics

- FID Scores → Measure image translation quality.
- Earth Mover’s Distance (EMD) → Assesses distribution transport accuracy.
- Translation Error (TE) → Evaluates content alignment precision.

Strengths:

- Clear synthetic data evaluations that visualize trajectory correctness.
- Shows that the bilevel loss improves identifiability beyond prior GAN/FM-based methods.

Weaknesses:

-  Table 1’s performance gains need further explanation → Whether the gains stem from translation identifiability or other factors is unclear.
- Table 2 (synthetic) provides stronger validation than Table 1 → More real-world experiments should match this clarity.
- Ablation studies are missing for:
(a) Alternative nonlinear interpolants.
(b) Computational efficiency trade-offs.
(c) Real-world experimental validation of the bilevel optimization loss's effect on translation identifiability.

**Other Comments Or Suggestions:**

Some mathematical derivations (e.g., Proposition 3.4) could be more clearly structured for better readability, and a summary table comparing DFM to existing methods would help contextualize its contributions.

**Other Strengths And Weaknesses:**

There are some weaknesses that could limit the paper’s broader impact.

- Empirical validation remains narrow, as the evaluation focuses on controlled synthetic settings and simple image translation tasks, without testing on real-world, high-dimensional structured datasets.

- Real-world validation of the bilevel loss is missing → Needs qualitative & quantitative ablation studies.

- Table 1 results require stronger interpretation to confirm that improvements arise from translation identifiability rather than other confounding factors.

- Computational trade-offs remain unexplored → Scalability to larger datasets needs further analysis.

- The lack of open-source implementation raises reproducibility concerns, which should be addressed before the camera-ready submission.


====

I will assess the authors' incorporation of feedback, along with other overall considerations, and determine whether to adjust or maintain the score accordingly.

**Questions For Authors:**

- What is the computational overhead of DFM compared to these baselines?

- Does the bilevel optimization framework introduce significant training inefficiencies?

- How sensitive is DFM’s performance to the specific choice of nonlinear interpolant?

- Have alternative nonlinear functions been tested, and if so, how do they affect stability and identifiability?

- Have you tested runtime performance across different dataset sizes or on high-dimensional data?

- Would the method perform well in cases where conditional distributions have overlapping supports, which may occur in real-world applications?

- Do you plan to release code before the camera-ready submission?

**Relation To Broader Scientific Literature:**

Additional Clarifications Needed:

- Clarify how DFM differs from prior work in nonlinear flow matching, particularly in handling mode overlaps and enforcing identifiability.
- More discussion on real-world applications where identifiability is a critical bottleneck.

**Theoretical Claims:**

I have no comments on that.

---

> ### Author Rebuttal · Authors · 2025-04-01
>
> &nbsp;
>
> **Anonymized URL**:  (https://drive.google.com/file/d/1U4gdB5qy1d98AJ1YWxR2v3fzDQp167nt/view?usp=sharing)
>
> &nbsp;
>
> **[Limitations of Claim 2: Bilevel Loss implies Identifiability]**
>
> To clarify, the proposed DFM and the baselines FM, FM-OT, and FM-cond do use the same architecture, and are trained using the same hyperparameter settings. Therefore, the only difference between these methods is in the loss design. Hence, the qualitative and quantitative performance gain can be attributed to the translation identifiability rather than architectural differences. We will add a remark for clarification.
>
> Regarding real-word scenario: in this work, our interest lies in a theoretically sound FM approach for realizing DDM and retaining translation identifiability. Translation identifiability’s effectiveness in real-world applications is indeed important, but a bit beyond the scope. Nonetheless, we noticed that there have been recent works using DDM and translation identifiability for medical imaging [R-1]. They did not use FM, but clearly showed the usefulness of translation identifiability. We will point readers to such references.
>
> We will make the above points clearer in the revised version.
>
> [R-1] Song, Jiahui, et al. "Translation Identifiability-Guided Unsupervised Cross-Platform Super-Resolution for OCT Images." IEEE SAM, 2024.
>
> &nbsp;
>
> **[Limitations of Claim 3: DFM achieves superior content alignment ]**
>
> We should mention that Table 1 is not meant to show performance improvement. Table 1 only shows that FID of the proposed method being comparable to the best baseline. This ensures that the image quality of the translated images is not compromised. Whereas, the qualitative results accompanying Table 1 in Fig. 7 is used to show that the proposed method has better content alignment than all other baselines. We have now quantified the content alignment result using DreamSim scores. This can be used together with Table 1 and the figure to assess the performance of the methods. Overall, a good method in our context should have low DreamSim scores and reasonable FID scores. Only having FID scores does not mean good performance.
>
> Note that we provided two real-world experiments with image translation and swarm navigation to show generalization beyond synthetic tasks.
>
> &nbsp;
>
> **[Table 1: Translation identifiability gains]**
>
> As mentioned, Table 1 is not used to show performance gain. Table 1 shows the image quality (measured by FID) is not compromised when we establish translation identifiability. The translation identifiability is shown in Fig. 7.
>
> The DFM does not retain identifiability if the nonlinear interpolants are not used. We have included linear interpolant in this experiment. One can see that the performance is not promising.
>
> &nbsp;
>
> **[Ablations]**
>
> **[Alternative non-linear Interpolants]** In fact, we did not use any specific design for the nonlinear interpolant. Our understanding is that as long as it satisfies the regularity condition and is a learnable interpolant, it can serve our purpose. To address the reviewer’s concern, we have conducted experiments on different nonlinear interpolant parametrization, which is shown in R-Fig. 1. In R-Fig. 1, we use two other parametrization of the nonlinear interpolant (i,e., $(1-t) x + ty + f(t) \gamma_{\theta}(x, y, t)$ with different $f(t)$, namely, (i) $ (1-t) x + t y + {\rm sin}(\pi t) \gamma_{\theta}(x, y, t)$ (ii) $ (1-t) x + t y + \sqrt{t (1-t) } \gamma_{\theta}(x, y, t)$. One can see that DFM is robust to different parametrization of the nonlinear interpolant.
>
>
> **[Computational efficiency]** R-Table 1 in the URL shows the training and inference time of all methods. The proposed method does not incur significant training time overhead compared to other FM based methods. Whereas, there is no overhead at all during inference.
>
>
> **[Real-world validation]** Note that our implementation did not directly optimize the bilevel loss. We exploited the non-overlapping structure of conditional distribution to obtain a simplified loss. We have provided two real-world validations as discussed previously.
>
> &nbsp;
>
> **[Overlapping supports]**
>
> R-Fig. 3 in the anonymized PDF link shows the result of overlapping supports of the conditional distribution. The figure shows that small overlaps do not significantly harm the translation. Further, note that non-overlapping supports naturally occur in applications as demonstrated in our experiments. That said, very large overlaps can indeed be detrimental to the proposed method.
>
> &nbsp;
>
> **[Code Submission]**
>
> Note that we have already provided demo code in the supplemental materials along with our manuscript. We will release the complete code along with pre-trained models before the camera-ready submission.

---

> > ### Comment · Reviewer_Y3HP · 2025-04-04
> >
> > Thank you for the author's response. However, it fell slightly short of my expectations, so I have adjusted the score to 3. While I acknowledge the study's contribution, I recommend that additional key experiments be conducted and included in the main paper to more effectively support and strengthen it.

---

> > > ### Author Response · Authors · 2025-04-05
> > >
> > > We thank the reviewer for the time and effort devoted to reviewing our work.
> > >
> > > We believe that we have addressed the reviewer’s concerns with further experiments or clarifications in our rebuttal (also, please see the attached PDF). These new experiments, along with clarifications, will be incorporated in the revised version.
> > > The additional experiments include:
> > > 1. Nonlinear interpolants,
> > >
> > > 2. Computational efficiency,
> > >
> > > 3. Overlapping supports,
> > >
> > > 4. More quantitative results for image translation (including DreamSim score and additional baselines).
> > >
> > > We have also addressed the following clarifications:
> > >
> > > 1. Table 1 (FID) should not be directly compared with Table 2. We use qualitative results, as shown in Fig. 7 of the manuscript, alongside the DreamSim score to assess translation performance. We will also expand the discussion of quantitative results, including responses to reviewer dKp3’s comments on the LPIPS score.
> > >
> > > 2. The image translation experiment is inherently high-dimensional and is a standard approach used in existing FM-based methods for validating real data.

---

### Official Review · Reviewer_wzYK · 2025-03-13

**Overall Recommendation:** 3

**Summary:**

The paper introduce Diversified Flow Matching (DFM), an FM-based framework for DDM. They design a custom loss function and nonlinear interpolant to ensure translation identifiability, addressing the limitations of conventional FM methods that use linear interpolants. By leveraging the non-overlapping property of conditional distributions, they reformulate the bilevel optimization problem into a two-stage approach, simplifying computation. The method is tested on synthetic data and real-world applications, demonstrating its effectiveness in maintaining translation identifiability and improving trajectory information.

**Claims And Evidence:**

YES

**Essential References Not Discussed:**

Deep Momentum Multi-Marginal Schrödinger Bridge:
The proposed DFM framework aims to unify multiple conditional distribution pairs under a single flow, which inherently aligns with multi-marginal SB formulations. Including this reference or related paper is critical to contextualize DFM’s bilevel optimization strategy against state-of-the-art SB advancements, especially since DFM implicitly addresses multi-marginal coupling.

**Experimental Designs Or Analyses:**

The paper evaluates DFM through three experiments: synthetic data, image translation, and swarm navigation. While the experiments demonstrate DFM’s effectiveness, several design and analysis choices warrant discussion:
1. Synthetic Data Experiment
Design:
Tests on 2D/3D Gaussian mixtures with ground-truth translation y=-x.
  Metrics: Earth Mover’s Distance (EMD) and Translation Error (TE).
Validity:
  Strengths: EMD and TE are appropriate for distribution matching and identifiability. The use of multiple trials (10 runs) adds statistical robustness.
2. Image Translation Experiment
Design:
  Translates CelebAHQ faces to Bitmoji using FID for distribution matching and visual checks for content alignment.
  Trains on Stable Diffusion’s latent space (not raw pixels).
Validity:
  Strengths: FID is standard for generative models. Including CycleGAN and DDM-GAN baselines contextualizes performance.
  Issues:
    Content Alignment: Relies on qualitative visual checks without quantitative metrics (e.g., SSIM, LPIPS, or user studies). This risks overlooking subtle misalignments.
3. Swarm Navigation Experiment
Design:
  Tests on LiDAR data of Mt. Rainier with Gaussian swarm sources/destinations.
  Metric: Surface Adherence (SA) to measure trajectory proximity to terrain.
Validity:
  Strengths: SA is a meaningful metric for terrain adherence.
  Issues:
    Oversimplified Scenario: Swarms have non-overlapping paths (per Assumption 3.5), ignoring real-world complexities like intersecting trajectories or dynamic obstacles.
General Issues:
Baseline Scope:
Omission of recent flow-based UDT methods (e.g., Rectified Flow, Schr̈odinger Bridges).

**Methods And Evaluation Criteria:**

The proposed methods and evaluation criteria in the paper are appropriate for the problem of diversified distribution matching (DDM).

**Other Comments Or Suggestions:**

No.

**Other Strengths And Weaknesses:**

Strengths:
1. Originality:
  The integration of FM with DDM is a novel contribution. While DDM was previously restricted to GANs, this work creatively adapts FM to enforce identifiability via bilevel optimization and learnable nonlinear interpolants.
  The design of private interpolants for each conditional distribution pair and the reformulation under non-overlapping structural assumptions demonstrate innovative problem-solving.
2. Significance:
  Translation identifiability is a critical issue in UDT, and DFM provides a principled solution with theoretical guarantees.
   The ability to recover transport trajectories has practical implications for applications like robot navigation and single-cell analysis, where path information is essential.
3. Technical Soundness:
   Theoretical results (e.g., Proposition 3.4) rigorously connect DFM to the DDM criterion, ensuring identifiability under sufficiently diverse conditional distributions.
   Experiments on synthetic data, image translation, and swarm navigation validate the method’s effectiveness. The improved FID scores (Table 1) and avoidance of GAN instability (Fig. 6) highlight empirical advantages.
4. Clarity:
  The paper is well-structured, with clear explanations of challenges in adapting FM for DDM (e.g., pitfalls of linear interpolants in Fig. 3). Visualizations (e.g., trajectories in Fig. 5, 9–11) enhance understanding.
Weaknesses:
1. Restrictive Assumptions:
  The efficient implementation relies on non-overlapping supports of conditional distributions (Assumption 3.5), which may not hold in many real-world scenarios. While acknowledged as a limitation, potential workarounds for overlapping cases are not explored.
2. Experimental Scope:
  Image translation experiments operate in a latent space rather than raw pixels, which may understate challenges in high-dimensional settings. Testing on more diverse modalities (e.g., text-to-image) could strengthen claims.

**Questions For Authors:**

No.

**Relation To Broader Scientific Literature:**

The key contributions of the paper are deeply rooted in addressing gaps and building upon advancements in unsupervised domain translation (UDT), distribution matching, and flow-based generative models. Here’s how they relate to prior work:
1. Bridging Flow Matching (FM) and Diversified Distribution Matching (DDM)
Prior Work:
  DDM (Shrestha & Fu, 2024): Introduced translation identifiability via GANs by matching multiple conditional distributions.
  Flow Matching (Lipman et al., 2022): Provided stable training and trajectory modeling via ODEs but focused on single distribution pairs.
Contribution:
  The paper unifies these frameworks by adapting FM to DDM, enabling translation identifiability (a GAN-based DDM strength) with trajectory information (a flow-based advantage). This resolves GANs’ instability and trajectory limitations while extending FM to multi-conditional settings.
2. Addressing Translation Non-Identifiability in Flow Models
Prior Work:
  CycleGAN (Zhu et al., 2017): Highlighted content misalignment due to non-unique transport maps.
  Optimal Transport (OT) in UDT (Liu et al., 2022; De Bortoli et al., 2021): Assumed OT maps as solutions but lacked guarantees for general translations.
Contribution:
  By enforcing DDM’s sufficiently diverse condition (SDC) through FM, the paper guarantees identifiability for non-OT maps. This directly addresses the non-uniqueness issue in prior UDT methods, aligning with theoretical insights from Shrestha & Fu (2024) but in a flow-based framework.
3. Introducing Bilevel Optimization and Nonlinear Interpolants
Prior Work:
  Linear Interpolants in FM (Albergo et al., 2023): Used linear paths (e.g., \(z_t=(1-t)x+ty\)) but failed for multi-conditional DDM due to conflicting trajectories.
  Conditional FM (Atanackovic et al., 2024): Learned separate flows per condition but lacked unified transport functions.
Contribution:
  The paper proposes nonlinear interpolants (Eq. 17) and bilevel optimization (Eq. 12) to harmonize trajectories across diverse conditional pairs. This extends FM’s applicability to DDM, resolving conflicts inherent to linear interpolants and enabling a unified velocity field.
4. Exploiting Structural Constraints for Efficiency
Prior Work:
  Non-Overlapping Supports in OT (Villani et al., 2009): Leveraged distribution separation for tractable transport.
Contribution:
  By assuming non-overlapping supports (Assumption 3.5) and designing non-intersecting interpolants (Eq. 14), the paper simplifies bilevel optimization into a tractable two-stage process. This aligns with structural insights from OT but tailors them to FM-based DDM.

**Theoretical Claims:**

The paper presents two main theoretical claims: Proposition 3.4 (translation identifiability of DFM) and Fact 3.3 (failure of the sum-of-LS formulation).
Proposition 3.4 relies on idealized assumptions (e.g., existence of diffeomorphisms) but is logically sound within the theoretical framework.
Fact 3.3 is rigorously supported by both synthetic examples and formal reasoning.

---

> ### Author Rebuttal · Authors · 2025-04-01
>
> &nbsp;
>
> **[Quantitative Metrics for Image to Image Translation]**
>
> Image similarity metrics were not presented because we observed that metrics such as LPIPS do not make sense when the images have large domain gaps (i.e., when the geometric representations of the feature spaces are largely different like in photos and bitmoji). For example, DFM and CycleGAN achieved the same LPIPS score, despite DFM being clearly better than CycleGAN in terms of alignment.
>
> Nonetheless, to alleviate the reviewer’s concern, we evaluate the DreamSim score [Fu et al, 2023]. This score serves for similar purposes of LPIPS but was shown to better align with human judgement [Fu et al., 2023]. Although the metric can provide some rough idea, it still cannot capture some obvious issues (e.g., DDM-GAN has diversity issues as seen in Fig. 7, last two rows, in the manuscript):
>
> ------------------------------
>
> Method    $\quad\quad$      Mean (Std)
>
> ------------------------------
>
> DFM       $\quad\quad\quad~~$        0.59 (0.05)
>
> DDM-GAN   $\quad$   0.58 (0.05)
>
> CycleGAN  $\quad~$    0.63 (0.06)
>
> FM   $\quad\quad\quad~~~~$  0.66 (0.06)
>
> FM-OT  $\quad\quad~~$       0.66 (0.06)
>
> FM-cond    $\quad~~$    0.60 (0.06)
>
> SDEdit	$\quad\quad~~$       0.62 (0.06)
>
> -------------------------------
>
> &nbsp;
>
> **[Swarm Navigation Issue]**
>
>
> We should mention that our learned paths are time-space nonoverlapping. This means that the swarms can use the overlapped space as long as they do not reach the same spatial point at the same time. This was actually reflected in Fig. 10 (**please note the colorbar**). We believe that such learned path plans are reasonable as they avoid collision and allow the swarms to use the same space at different times. We realize that we did not articulate this point in the simulations. We will clarify in the revision.
>
> &nbsp;
>
> **[Additional Baselines]**
>
> **Anonymized URL**: (https://drive.google.com/file/d/1U4gdB5qy1d98AJ1YWxR2v3fzDQp167nt/view?usp=sharing)
>
> To alleviate the reviewer’s concern, we have conducted experiments with additional baseline SDEdit which is based on diffusion. The new table and figure can be found in the anonymized pdf link. One can see that SDEdit suffers from misalignment as well as has higher FID.
>
> &nbsp;
>
> **[References]**
>
> Thank you for the relevant reference. We will discuss this work in our related work section.
>
> &nbsp;
>
> **[Non-overlapping support assumption]**
>
> In fact, we found that the non-overlapping support assumption is not very hard to be met in practice, as the partitioning according to $u^{(q)}$ is controlled by the system designers. For example, in photo to cartoon translation, one can pick black/non-black hairs as $u^1$ and $u^2$, which naturally lead to non-overlapping clusters. Of course, partitioning data would reduce the amount of data in each cluster, which might make learning its corresponding $v_t$ harder. This is a tradeoff that system designers should pay attention to when splitting the data.
>
> &nbsp;
>
> **[Experimental Scope]**
>
> The reviewer’s comment is related to an open challenge for diffusion and FM-type methods; that is, scalability with large dimension data has not been solved in general. Note that diffusion and flow matching in latent space of VAE rather than raw pixels is common practice [R1, R2] . Further, the latent space is of dimension 32 x 32 x 4, which is still quite high dimensional.
>
> We agree that diverse modalities like text-image is a very interesting scenario. However, applying any diffusion/FM models in the raw data space remains an open problem, and solving this challenge is out of scope of this work. We will add some remarks in “limitations” to draw attention to this open challenge.
>
> *[R1] Kapusniak, Kacper, et al. "Metric flow matching for smooth interpolations on the data manifold." NeurIPS 2024.*
>
> *[R2] Tong, Alexander, et al. "Improving and generalizing flow-based generative models with minibatch optimal transport.*

---

### Official Review · Reviewer_AAms · 2025-03-14

**Overall Recommendation:** 2

**Summary:**

This paper aims to address the unpaired domain translation problem with conditional information. The previous method is based on GAN. However, GAN training may not be stable; this paper proposes a Flow Matching-based method. However, the naive FM method may not work because conditional distributions may be mismatched, as shown in the synthetic experiments in the paper. The authors proposed a novel bilevel optimization loss to address this problem. One level of the loss optimizes the flow for each condition, and the other level optimizes the global flow and the interpolant between source and target distributions. The authors conducted experiments on synthetic data, Human face data, and Swarm Navigation data.

**Claims And Evidence:**

Yes.

**Essential References Not Discussed:**

Several existing image-to-image translation methods are missing:
- Zhao et al, EGSDE: Unpaired Image-to-Image Translation via Energy-Guided Stochastic Differential Equations, NeurIPS 2022.
- Gazdieva et al., Extremal Domain Translation with Neural Optimal Transport, NeurIPS 2023.
- Kornilov et al., Optimal Flow Matching: Learning Straight Trajectories in Just One Step. NeurIPS 2024.

**Experimental Designs Or Analyses:**

Yes, important baselines are missing, and experiments on more datasets on which domain translation methods are commonly tested should be included.

**Methods And Evaluation Criteria:**

Not sufficient. The toy examples should include the case that different conditions have overlaps. The authors need to compare with stronger baselines on more datasets.

**Other Comments Or Suggestions:**

No.

**Other Strengths And Weaknesses:**

### Strength:
The authors proposed a Flow Matching method to address the unpaired domain translation problem with conditional information. The authors proposed a bilevel optimization loss. One level of the loss optimizes the flow for each condition, and the other level optimizes the global flow and the Interpolant between source and target distributions.

### Weakness:

The experiments in this paper are not strong.
1. The authors should compare with stronger baselines:
- Zhao et al, EGSDE: Unpaired Image-to-Image Translation via Energy-Guided Stochastic Differential Equations, NeurIPS 2022.
- Gazdieva et al., Extremal Domain Translation with Neural Optimal Transport, NeurIPS 2023.
- Kornilov et al., Optimal Flow Matching: Learning Straight Trajectories in Just One Step. NeurIPS 2024.

2. The authors should do experiments on more datasets, such as CelebA/CelebA-HQ, AFHQ, on which domain translation methods are commonly tested.

3. In the synthetic experiments, the authors should include data with overlapping condition distributions, such that we can evaluate how effective Eq. 12 is.

4. The improvement of the proposed method is minor compared to using the naive baseline FM-cond: 22.21 vs 22.65 in terms of FID.

**Questions For Authors:**

In Fig. 3 (b), the authors drew intended trajectories. However, I am not sure whether the blue and green trajectories are really achievable or not. The blue and red lines cross. At each cross point of one blue line and one red line, the final velocity v will be the average of the two velocities of the red and green lines. So, the blue and red lines cannot cross on the 2-d plane. I have similar concerns for Fig. 5 (a) and Fig. 10 DFM. The authors need to clarify this in case my understanding is wrong.

The authors did synthetic experiments showing Eq. 10 sometimes works and sometimes doesn't work, to motivate their own formulation Eq. 12. However, the authors didn't explain why Eq. 10 doesn't work. Is the network complexity not enough, or is the initialization not good, or other reasons? How do you parametrize $I^{(q)}$ in Eq. 10 and 12? Eq. 12 is also difficult to optimize, and Eq. 12 could also sometimes fail. The authors need to justify that the optimal solution of Eq. 12 can be achieved in the experiments.

**Relation To Broader Scientific Literature:**

The key contribution of this paper is to propose a Flow Matching method to address the domain translation problem with additional conditional information. The previous method proposed by Shrestha and Fu 2024 was based on GAN.

**Theoretical Claims:**

I roughly checked the proof of Proposition 3.4, and didn't find issues.

---

> ### Author Rebuttal · Authors · 2025-04-01
>
> **[Intersecting trajectories in figures]**
>
> Note that Fig. 3(b) does **not** show the velocity field but the interpolant $I^{\rm linear}(x, y, t)$ where $x, y \sim \rho(x, y | u^{(q)})$. The wording was meant to imply that interpolants guide the learning of the vector field. Our term “intended trajectories” could be confusing; we will change it to “interpolant trajectories.”
>
> In Fig. 5(a) and Fig. 10, the trajectories do not cross each other (**please note the time colorbar**). This is because the two conditional populations travel at different speeds. Hence the intermediate transported distributions are not at the same location at the same time, although they pass via the same location, avoiding collision. This was briefly discussed in Sec. 5.1 (2D Gaussian blobs). We will make this clearer in the revised version.
>
> &nbsp;
>
> **[About Eq (10) and (12)]**
>
> This is an important but nuanced point worth clarifying. We briefly discussed why Eq. (10) fails in Lines 244–268, with a more rigorous explanation in the proof of Fact 3.3. Essentially, when $I^{(q)}$ is learnable and $Q > 1$, Eq. (11) may not hold for all $q$, even though it is needed for $\widehat{v}$ to transport $p_{x|u^{(q)}}$ to $p_{y|u^{(q)}}$. Changing $I^{(q)}$ alters the minimum of the flow matching loss $L_{\rm FM}^{(q)}$, and since $v_t$ is coupled with all $L_{\rm FM}^{(q)}$, minimizing Eq. (10) may not achieve the minimum for every loss. However, since $v_t$ is coupled with all $L_{\rm FM}^{(q)}$’s, it is not guaranteed that minimum of all $L_{\rm FM}^{(q)}$ is reached by minimizing Eq. (10).
>
> $I^{(q)}$ is also parameterized as in Eq. (17).
>
> Note that Eq. (12) does not “fail” like Eq. (10); unlike Eq. (10), which is theoretically flawed (therefore fails sometimes) as shown in Fact 3.3, Eq. (12) is theoretically sound per Prop. 3.4. However, as the reviewer correctly noted, Eq. (12) is hard to optimize due to its bilevel nature. To address this, we proposed to exploit the structural constraint in Section 3.4.
> We will add a remark to summarize these nuances and clarify this discussion.
>
> &nbsp;
>
> **[More Experiments]**
>
> **Anonymized URL**: (https://drive.google.com/file/d/1U4gdB5qy1d98AJ1YWxR2v3fzDQp167nt/view?usp=sharing)
>
> We should mention that the baselines FM-OT and MFM are strong, recent methods. Nonetheless, to address the reviewer’s concerns, we experimented with two additional diffusion-based baselines—EGSDE (provided by the reviewer) and SDEdit [Meng et al., 2022]—both using the same diffusion model trained on the Bitmoji domain based on DDIM.
>
> In URL, we include these baselines and test them on the CelebAHQ-to-Bitmoji experiment (note that standard datasets, e.g., CelebAHQ Male-to-Female and AFHQ, are actually easier cases since the domain gap is smaller; hence, content misalignment rarely appears even for methods (such as MFM, EGSDE) without identifiability guarantees).
>
> R-Fig. 3 shows that translations by the new baselines differ significantly from Bitmoji images, leading to high FID despite setting the reverse diffusion repeats (K) to 3. Thus, although the ^DreamSim score [Fu et al., 2023] is relatively low, the translations are not meaningful due to high FID. In comparison, our method yields superior translations, as shown by qualitative results, FID, and DreamSim scores.
>
> *^Note that DreamSim appears to be insensitive to small but perceptible differences in translation fidelity (c.f. DFM and DDM-GAN qualitative results)*
>
> &nbsp;
>
> **[References]**
>
> Thank you for the relevant references. We will include them in the revised version.
>
> &nbsp;
>
> **[Overlapping Cases]**
>
> R-Fig. 3 shows the case where the source and target conditional distributions have some overlap. The overlap is created by increasing the variance of the Gaussian components from Fig. 3 in the manuscript. One can see that small overlaps have a marginal effect on the performance of the proposed method. However, since the proposed implementation exploits the property of disjoint supports, larger overlaps can be detrimental to its performance.
>
> We should remake that, as the partitioning of the conditional distribution is **controlled by the system designer,** the available data can always be split in a support-disjoint way. Hence, the harm brought by overlapping support is avoidable in many cases.
>
> &nbsp;
>
> **[About FID]**
>
> **In the context of identifiability-guaranteed translation, a good method retains source content while maintaining a reasonable FID (image quality).** Thus, minimizing FID is not our primary goal; we use it to show that our method does not compromise quality. Note that low FID alone does not ensure transport identifiability. Even if some baselines have similar FID scores, it does not mean they could attain content alignment.
>
> R-Table 1 shows that our method does not compromise image quality (measured by FID) while ensuring content alignment (see R-Fig. 2 and Dreamsim score), unlike baselines that often lose content despite similar FID scores.

---

> > ### Comment · Reviewer_AAms · 2025-04-05
> >
> > Thank the authors for the response! However, the improvement of the proposed method is still marginal compared to the FM-cond from results in the main paper and the additional results in the rebuttal regarding the Dreamsim score (0.59 vs 0.60). Also, more stronger baselines (e.g. Kornilov et al. 2024) I mentioned in the review were not compared, more datasets and standard domain translation tasks were not tested. I would still consider experiment part a week point of this paper.

---

> > > ### Author Response · Authors · 2025-04-05
> > >
> > > Thank you for the time and effort devoted to reviewing our work.
> > > We believe that we have addressed these concerns in our rebuttal. To further clarify:
> > >
> > > 1. We selected a baseline from the list provided by the reviewer, identified as one of the “strong baselines.” Kornilov et al., 2024, was not chosen because it requires training ALAE before the flow matching can be trained—a process that would be challenging to complete within the rebuttal period given our current resource and time constraints. We do plan to include it in the revised version.
> > >
> > > 2. We acknowledge that the Dreamsim score offers only a rough indication of performance. It does not capture several nuances in translation quality (for example, DDM-GAN has a lower Dreamsim score but exhibits noticeable translation and diversity issues compared to DFM). We have provided additional explanations in our response to Reviewer dKp3 under the [LPIPS score] section, and we will add more qualitative results to the appendix in the revised version.
> > >
> > > 3. We provided clarifications regarding standard domain translation tasks such as CelebA-HQ male-to-female, where existing methods are already capable of producing content-aligned translations. Our method specifically addresses content-misalignment issues observed in other approaches. However, it is important to note that standard datasets (e.g., male-to-female, AFHQ) do not exhibit these issues to begin with.
> > >
> > > We hope these explanations clarify our approach and thank you for your valuable feedback.

---

### Decision · Program_Chairs · 2025-05-01

**Decision:**

Accept (poster)

**Comment:**

The reviewers generally appreciated the theoretic development and the integrating of prior research. Particularly, the integration of translation identifiability via Diversified Distribution Matching (DDM) and flow matching (FM) was interesting. The theoretic development seems to be the main strength of the paper.

However, the original goal for merging DDM and FM is to provide an empirically better UDT method that preserves content better. But, as multiple reviewers noted, the experiments do not seem to clearly outperform previous methods (particularly when the compared baselines are stronger). Some noted that the original experiments did not include strong baselines.

Additionally, the main claim of the paper relies on qualitative assessment of content/semantic preservation. But the qualitative results (especially with the stronger baselines) are not particularly convincing. By looking at the new results with the added baselines, it is not clear that DFM outperforms them empirically (e.g., see rows 3 and 4 in R-Fig. 2 where EGSDE arguably performs better, DFM seems to perform better for rows 1 and 2). Thus, the key benefit of the proposed method is not clearly demonstrated.

While theoretic identifiability is interesting in its own right, this paper builds upon the prior identifiability results. And thus, the significance of the proposed approach hinges on better empirical performance.

Given this, I think this is an interesting work but the experiments need strengthening and clarifying to more clearly demonstrate the benefits of the theoretically developed methods.